# Interleaving Multi-Task Neural Architecture Search

## Abstract

Multi-task neural architecture search (MTNAS), which searches for a shared architecture for multiple tasks, has been broadly investigated. In these methods, multiple tasks are learned simultaneously by minimizing the weighted sum of their losses. How to balance these losses by finding the optimal loss weights requires a lot of tuning, which is time-consuming and labor intensive. To address this problem, we propose an interleaving MTNAS framework, where no tuning of loss weights is needed. In our method, a set of tasks (e.g., A, B, C) are performed in an interleaving loop (e.g., ABCABCABC...) where each task transfers its knowledge to the next task. Each task is learned by minimizing its loss function alone, without intervening with losses of other tasks. Loss functions of individual tasks are organized into a multi-level optimization framework which enables all tasks performed end-to-end. The effectiveness of our method is demonstrated in a variety of experiments.

## 1 Introduction

Neural architecture search (Zoph & Le, 2017; Liu et al., 2019a; Real et al., 2019), which aims at automatically searching for high-performance neural architectures with minimal human intervention, has attracted much attention recently and demonstrated promising effectiveness in a variety of computer vision tasks, including classification, segmentation, object detection, image captioning, etc. When multiple tasks are closely related, it is beneficial to search for a shared architecture for all tasks via multi-task learning. A number of methods have been developed for multi-task NAS (MTNAS) (Liang et al., 2018; Bruggemann et al., 2020; Gao et al., 2020). Existing MTNAS methods learn a shared architecture by minimizing the weighted sum of losses of all tasks. How to balance these loss terms is challenging: typically, more decrease of one loss renders less decrease of other losses; as a result, improving one task renders performance degradation of other tasks. To alleviate this problem, it is needed to carefully tune the tradeoff weights between loss terms (please see Table 4 for empirical justifications), which is time-consuming and labor-intensive.

We aim to address this problem, by proposing an interleaving multi-task NAS framework. Different from existing MTNAS methods which perform multiple tasks by minimizing a combined loss function, our method is featured with multiple loss functions where each loss focuses on learning one task. This can avoid tuning the tradeoff weights between losses of different tasks. Different tasks are organized into a loop where each task can transfer its knowledge to any other task along the loop. Loss functions of individual tasks are unified into a multi-level optimization framework (Franceschi et al., 2018) which enables different tasks to be performed end-to-end.

In our framework, there are $K$ tasks whose learnable parameters include network weights and a shared architecture. Each task has a data encoder and a task-specific head. Data encoders of all tasks share the same architecture, but have different network weights. The $K$ tasks perform $M$ rounds of interleaving learning. In the first round, we first learn task $t_1$, then learn task $t_2$, and so on. At the end of the first round, $t_K$ is learned. Then we move to the second round, which starts with learning $t_1$, then learns $t_2$, and so on. This pattern repeats until the $M$ rounds of learning are finished. Between two consecutive tasks $t_k$ and $t_{k+1}$, knowledge transfer is performed based on distribution matching. Since all tasks are in a loop, the knowledge of one task can be propagated to any other task. After $M$ rounds of learning, each task uses its model trained in the final round to make predictions on a validation dataset and updates their shared architecture by minimizing validation losses.

The major contributions of this paper are as follows:

- We propose an interleaving multi-task neural architecture search method, which searches for a shared architecture of multiple tasks in an interleaving fashion. Our framework solves multiple optimization problems

sequentially, each focusing on training one task, which therefore can circumvent the difficulty of balancing different loss terms.

- We formulate interleaving MTNAS as a multi-level optimization problem which enables all tasks to be performed end-to-end.

- We propose a new knowledge transfer method based on distribution matching.

- Experiments on image classification and object detection demonstrate the effectiveness of our method.

## 2 Related works

### 2.1 Multi-task learning (MTL)

MTL (Ruder, 2017; Zhang & Yang, 2021) aims to improve multiple tasks simultaneously by learning them jointly and transferring knowledge across tasks. Various MTL approaches have been proposed, based on 1) hard parameter sharing (Caruana, 1998; Long et al., 2015; Doersch & Zisserman, 2017; Kokkinos, 2017; Sener & Koltun, 2018; Leang et al., 2020), where multiple tasks share the same weight parameters, such as encoder weights; 2) soft parameter sharing (Duong et al., 2015; Dai et al., 2016; Misra et al., 2016; Yang & Hospedales, 2016; Lu et al., 2017; Liu et al., 2019b; Maninis et al., 2019), where parameters of different tasks are constrained to be similar; 3) task similarity learning (Williams et al., 2007; Zhang & Yeung, 2014; Bingel & Søgaard, 2017; Standley et al., 2020; Zamir et al., 2020), which identifies similarity between tasks and encourages similar tasks to share more commonalities; 4) loss weighting (Chen et al., 2018; Kendall et al., 2018; Sener & Koltun, 2018; Gong et al., 2019; Leang et al., 2020), which weighs each task's loss. Existing methods mostly learn multiple tasks by minimizing the weighted sum of their losses, which requires careful tuning of tradeoff weights for balancing different loss terms. Different from these methods, our method performs MTL in an interleaving way. Zeng et al. (2019) proposed a cyclic MTL approach for neural simile recognition where several subtasks are learned in a loop. Tian et al. (2021) proposed to cyclically co-learn sounding object visual grounding and audio-visual sound separation. In these two approaches, subtasks are learned by minimizing the weighted sum of their training losses, which requires extensive tuning of tradeoff weights as well. In an audio source separation task, Doire & Okubadejo (2019) minimize the sum of losses of different sources by alternating mini-batches of data from different sources. This work suffers the difficulty of balancing loss terms since the mini-batch update of one source may lead to loss increase of other sources.

### 2.2 Neural architecture search (NAS)

The goal of neural architecture search (NAS) is to automatically identify highly-performing neural architectures that can potentially surpass human-designed ones. NAS research has made considerable progress in the past few years. Early NAS (Zoph & Le, 2017; Pham et al., 2018; Zoph et al., 2018) approaches are based on reinforcement learning (RL), where a policy network learns to generate high-quality architectures by maximizing the validation accuracy (as reward). These approaches are conceptually simple and can flexibly perform search in any search space. In differentiable search methods (Cai et al., 2019; Liu et al., 2019a; Xie et al., 2019), each candidate architecture is a combination of many building blocks. The combination coefficients represent the importance of building blocks. Architecture search amounts to learning these differentiable coefficients, which can be done using differentiable optimization algorithms such as gradient descent. Differentiable NAS methods started with DARTS (Liu et al., 2019a) and have been improved rapidly since then. For example, P-DARTS (Chen et al., 2019) allows the architecture depth to increase progressively during searching. It also performs search space regularization and approximation to improve stability of searching algorithms and reduce search cost. In PC-DARTS (Xu et al., 2020), the redundancy of search space exploration is reduced by sampling sub-networks from a super network. It also performs operation search in a subset of channels via bypassing the held-out subset in a shortcut. Another paradigm of NAS methods (Liu et al., 2018b; Real et al., 2019) are based on evolutionary algorithms (EA). In these approaches, architectures are considered as individuals in a population. Each architecture is associated with a fitness score representing how good this architecture is. Architectures with higher fitness scores have higher odds of generating offspring (new architectures), which replace architectures that have low-fitness scores. Our framework is orthogonal to existing NAS approaches and is applicable to most differentiable methods.

There are several works proposed for multi-task neural architecture search. Liang et al. (2018) proposed an evolutionary algorithm based multi-task NAS method which searches for a routing of modules for each task where candidate modules are shared by different tasks. Pasunuru & Bansal (2019) developed a framework for multi-task and continual NAS. Bruggemann et al. (2020) proposed branched multi-task architecture search, to automatically determine encoder branching in multi-task architectures. Gao et al. (2020) proposed a task-agnostic NAS framework for general-purpose multi-task learning. Silvestri (2020) proposed a one-shot NAS method for multi-task learning. Cai & Luo (2021) proposed a multi-task learning framework for conducting multi-objective evolutionary NAS. Similar to the MTL methods discussed in Section 2.1, these multi-task NAS methods require extensive tuning of tradeoff weights as well.

### 2.3 Bi-level and multi-level optimization

A bi-level optimization (BLO) problem has two nested levels of optimization problems. At the lower level, model weights are learned by minimizing a training loss. At the upper level, meta parameters are learned by minimizing a validation loss. BLO has been applied for neural architecture search (Liu et al., 2019a; Zhang et al., 2021), hyperparameter optimization (Baydin et al., 2018; Feurer et al., 2015; Franceschi et al., 2017; 2018; Lorraine et al., 2020; Maclaurin et al., 2015), reinforcement learning (Hong et al., 2020a; Konda & Tsitsiklis, 1999; Rajeswaran et al., 2020), data valuation (Ren et al., 2020; Shu et al., 2019; Wang et al., 2020b), meta learning (Finn et al., 2017; Rajeswaran et al., 2019), and label correction (Zheng et al., 2019) where meta parameters are neural architectures, hyperparameters, importance weights of data, etc. Many optimization algorithms (Couellan & Wang, 2016; Ghadimi & Wang, 2018; Grazzi et al., 2020; Ji et al., 2021; Liu et al., 2021; Yang et al., 2021) have been proposed for solving BLO problems. Multi-level optimization (MLO) (Garg et al., 2022; He et al., 2021; Raghu et al., 2021; Somayajula et al., 2022; Such et al., 2020; Xie & Du, 2022) is an extension of BLO, with multiple levels of nested optimization problems. MLO has been applied to formulate problems that involve multiple learning stages which need to be conducted end-to-end. Sato et al. (2021) proposed a gradient-based optimization algorithm for solving MLO problems with theoretical guarantees.

### 2.4 Transfer learning (TL)

Our work is also related to TL (Pratt, 1993; Mihalkova et al., 2007; Niculescu-Mizil & Caruana, 2007; Pan & Yang, 2009; Luo et al., 2017; Zhuang et al., 2020), which aims to improve performance of a target task by transferring knowledge from a source task. Various transfer strategies have been proposed, based on reweighting source data to better match target distribution (Huang et al., 2006; Jiang & Zhai, 2007; Sugiyama et al., 2007; Foster et al., 2010; Moore & Lewis, 2010; Axelrod et al., 2011; Wang et al., 2017b; Ngiam et al., 2018), transforming source and target data into a common feature space (Borgwardt et al., 2006; Pan et al., 2008; 2010; Duan et al., 2012; Long et al., 2013; Wang et al., 2017a), extracting domain-invariant representations based on adversarial learning (Ganin et al., 2016; Long et al., 2017; Tzeng et al., 2017; Hoffman et al., 2018; Shen et al., 2018; Zhang et al., 2019), regularizing target model using source model (Luo et al., 2008; Duan et al., 2009; Zhuang et al., 2009; Tommasi et al., 2010; Duan et al., 2012), etc. Unlike TL which focuses on learning a target task with a source task as auxiliary, MTL methods (including ours) aim to improve all tasks and have no source/target distinction.

Our work is also related to meta learning, which learns to transfer meta knowledge to end tasks (Andrychowicz et al., 2016; Li & Malik, 2016; Ravi & Larochelle, 2016; Finn et al., 2017; Li et al., 2017; Mishra et al., 2017; Antoniou et al., 2018; Gidaris & Komodakis, 2018; Lee & Choi, 2018; Qiao et al., 2018; Hospedales et al., 2020; Lekkala & Itti, 2020; Sun et al., 2020; Yao et al., 2020). Different from these methods, our method transfers knowledge directly between end tasks.

## 3 Method

In this section, we present details of the interleaving MTNAS framework. There are $K$ tasks. These tasks could be the same (e.g., image classification on CIFAR-10, on CIFAR-100) or different (e.g., image classification on CIFAR-10, object detection on MS-COCO (Lin et al., 2014)). Each task $k$ has a deep neural network model, consisting of a data encoder and a task-specific head. Given an input data example, it is fed into the encoder to generate a latent representation; the latent representation is fed into the head to predict an output label. Each task $k$ has a training dataset $D_k^{(\text{tr})}$ and a validation dataset $D_k^{(\text{val})}$. The weight parameters of an encoder and head can be different in different rounds.

Let $W_k^{(m)}$ and $H_k^{(m)}$ denote the encoder weights and head weights of task $k$ in round $m$. Besides weight parameters, these tasks have a shared architecture $A$ to learn. These tasks perform interleaving learning (with $M$ rounds) in the following order:

$$\underbrace{t_1, t_2, \cdots, t_K}_{\text{Round 1}} \underbrace{t_1, t_2, \cdots, t_K}_{\text{Round 2}} \cdots \underbrace{t_1, t_2, \cdots, t_K}_{\text{Round } m} \cdots \underbrace{t_1, t_2, \cdots, t_K}_{\text{Round } M} \tag{1}$$

where $t_k$ denotes that the $k$-th task performs learning. In the first round, $t_1$ is learned first, then $t_2$ is learned, then $t_3$, etc. When learning $t_{k+1}$, we transfer knowledge from $t_k$ to $t_{k+1}$. After all $K$ tasks are learned in round 1, we move to the next round. When learning $t_1$ in the second round, we transfer knowledge from $t_K$ learned in the first round to $t_1$. Then $t_2$ is learned using knowledge transferred from $t_1$, and so on. This interleaving procedure iterates for $M$ rounds. Table 1 shows the notations of our method.

## 3.1 A multi-level optimization framework

There are $M \times K + 1$ learning stages: in each of the $M$ rounds, each of the $K$ tasks is learned at a stage; additionally, there is a final validation stage.

### 3.1.1 The first stage

At the very first stage, we train the model weights of task 1 in round 1, including encoder weights $W_1^{(1)}$ and task head $H_1^{(1)}$, by minimizing a training loss $L_{1,1}^{(tr)}(\cdot)$. The loss is defined on the model of task 1 in round 1 (including $W_1^{(1)}$, $H_1^{(1)}$, architecture $A$) and the training data $D_1^{(\text{tr})}$ of task 1. $L_{1,1}^{(tr)}(\cdot)$ is application specific. For example, in classification, $L_{1,1}^{(tr)}$ is a cross-entropy loss; given

Table 1: Notations in our method

| Notation | Meaning |
|---|---|
| $K$ | the number of tasks |
| $D_k^{(\text{tr})}$ | the training dataset of task $k$ |
| $D_k^{(\text{val})}$ | the validation dataset of task $k$ |
| $W_k^{(m)}$ | the encoder weights of task $k$ in round $m$ |
| $H_k^{(m)}$ | the head weights of task $k$ in round $m$ |
| $A$ | meta parameters |
| $M$ | the number of rounds |
| $K$ | the number of tasks |
| $C$ | the number of augmented data examples |
| $L_{k,m}^{(tr)}$ | training loss defined on the model of task $k$ in round $m$ |
| $L_k^{(val)}$ | validation loss of task $k$ |

a data-label pair $(x, y)$, the input data $x$ is first fed into $W_1^{(1)}$ then $H_1^{(1)}$ to yield a prediction $\hat{y}$, and cross-entropy loss is defined on $\hat{y}$ and ground-truth label $y$. $A$ is needed to define $L_{1,1}^{(tr)}$, but is not optimized at this stage. Otherwise, a trivial solution of $A$ will be yielded where $A$ can perfectly overfit $D_1^{(\text{tr})}$ but has poor generalization capability. The optimization problem is:

$$\widetilde{W}_1^{(1)}(A), \widetilde{H}_1^{(1)}(A) = \operatorname{argmin}_{W_1^{(1)}, H_1^{(1)}} L_{1,1}^{(tr)}(\cdot)(A, W_1^{(1)}, H_1^{(1)}, D_1^{(\text{tr})}) \tag{2}$$

$\widetilde{W}_1^{(1)}(A)$ denotes that the optimal weights $\widetilde{W}_1^{(1)}$ depend on $A$ since 1) the training loss is a function of $A$ and 2) $\widetilde{W}_1$ depends on the training loss.

### 3.1.2 A middle stage

At any other learning stage, e.g., the $k$-th stage in the $m$-th round, we learn the model weights of the $k$-th task in the $m$-th round, including encoder weights $W_k^{(m)}$ and task head $H_k^{(m)}$, by solving two optimization problems. In the first problem, we train $W_k^{(m)}$ by transferring knowledge from $\widetilde{W}_{k-1}^{(m)}(A)$. Previous knowledge transfer approaches (Rajeswaran et al., 2019; Romero et al., 2014; Chen et al., 2020a) have two major limitations. First, they impose strong restrictions on model parameters (Rajeswaran et al., 2019) and data embeddings (Romero et al., 2014), for example, making the weight parameters of two models have a small L2 distance (Rajeswaran et al., 2019). Second, they are limited to capturing low-order (e.g., pairwise (Chen et al., 2020a)) relationships between data examples, which cannot capture the global properties of a dataset sufficiently and therefore hinders knowledge transfer through this dataset. To address the limitations of existing methods, we propose a new knowledge transfer approach based on distribution matching, which offers two benefits. First, it is more flexible in the sense that it does not impose strong restrictions that weights or embeddings need to be close in terms of L2 distance as previous methods do. Second, it measures

similarity at the distribution level instead of individual example level, therefore can capture high-order ($> 3$) relationships between data examples and more holistic properties of an entire data distribution, which facilitates more effective knowledge transfer.

We generate sets of examples where examples in the same set are from the same distribution, then compare the distributions of two example-sets and see whether they are the same. To create a set of examples that are from the same distribution, we resort to data augmentation, based on the intuition that augmented examples created from the same original example can be considered as being from the same distribution. From each input training example $x_i$ (excluding its output label) in task $k$ and $k-1$, data augmentation is applied to it to generate a set of $C$ augmented examples $\mathcal{A}_i = \{a_n^{(i)}\}_{n=1}^C$, which are considered as samples drawn from the same distribution since they all originate from $x_i$.

Given two augmentation sets $\mathcal{A}_i$ and $\mathcal{A}_j$, we encourage $W_k^{(m)}$ and $\widetilde{W}_{k-1}^{(m)}(A)$ to be consistent in predicting whether the distributions of $\mathcal{A}_i$ and $\mathcal{A}_j$ are the same. We first use the optimal weights $\widetilde{W}_{k-1}^{(m)}$ of $t_{k-1}$ to label whether $\mathcal{A}_i$ and $\mathcal{A}_j$ are from the same distribution in the following way. We use $\widetilde{W}_{k-1}^{(m)}$ to encode examples in $\mathcal{A}_i$ and $\mathcal{A}_j$, and measure the maximum mean discrepancy (MMD) (Gretton et al., 2012) on the encodings. Let $z_n^{(i)}$ denote the embedding of $a_n^{(i)}$ by $\widetilde{W}_{k-1}^{(m)}$ and $f(\cdot, \cdot)$ be a kernel function. The MMD between $\mathcal{A}_i$ and $\mathcal{A}_j$ can be calculated as:

$$\frac{1}{C(C-1)} \sum_{n=1}^C \sum_{m \neq n}^C f(z_n^{(i)}, z_m^{(i)}) + \frac{1}{C(C-1)} \sum_{n=1}^C \sum_{m \neq n}^C f(z_n^{(j)}, z_m^{(j)}) - \frac{2}{C^2} \sum_{n=1}^C \sum_{m=1}^C f(z_n^{(i)}, z_m^{(j)}). \tag{3}$$

If the MMD is less than a threshold $\tau$, $\mathcal{A}_i$ and $\mathcal{A}_j$ are considered to be from the same distribution.

Given these binary labels (regarding whether two augmentation sets are from the same distribution), we learn $W_k^{(m)}$ of $t_k$ by fitting these labels. Specifically, we use $W_k^{(m)}$ to embed $\mathcal{A}_i$ and $\mathcal{A}_j$, and calculate the MMD $d(\mathcal{A}_i, \mathcal{A}_j; W_k^{(m)})$. If $\mathcal{A}_i$ and $\mathcal{A}_j$ are labeled as being from the same distribution by $\widetilde{W}_{k-1}^{(m)}$, then we encourage $d(\mathcal{A}_i, \mathcal{A}_j; W_k^{(m)})$ to be less than $\tau$. We use hinge loss to encourage the MMD calculated by $W_k^{(m)}$ and $\widetilde{W}_{k-1}^{(m)}(A)$ to be consistent: if they are both larger than $\tau$ or smaller than $\tau$, there are no penalty. The optimization problem is:

$$\widetilde{W}_k^{(m)}(A) = \mathrm{argmin}_{W_k^{(m)}} \sum_{\mathcal{A}_i, \mathcal{A}_j} \max\left(0, -\big(d(\mathcal{A}_i, \mathcal{A}_j; \widetilde{W}_{k-1}^{(m)}(A)) - \tau\big)\big(d(\mathcal{A}_i, \mathcal{A}_j; W_k^{(m)}) - \tau\big)\right), \tag{4}$$

where the loss is defined on pairs of augmentation sets. In practice, we randomly sample a small number (specifically, 20) of input training examples to generate augmentation sets. Half of these examples are from task $k-1$ and the other half are from task $k$. The computation cost on the 20 sets is not high. When $k = 1$ and $m > 1$ (at the beginning of a round), the optimal encoder weights of the previous task is $\widetilde{W}_K^{(m-1)}(A)$ (at the end of previous round). The optimal encoder weights $\widetilde{W}_k^{(m)}$ depend on $A$. $A$ is not updated at this learning stage, for the same reason described above.

In the second optimization problem, given the trained encoder $\widetilde{W}_k^{(m)}(A)$, we train the head $H_k^{(m)}$ by minimizing a training loss $L_{k,m}^{(tr)}$ defined on the model of task $k$ in round $m$ (including $\widetilde{W}_k^{(m)}(A)$, $H_k^{(m)}$, architecture $A$) and the training data $D_k^{(\mathrm{tr})}$ of task $k$. The second optimization problem is:

$$\widetilde{H}_k^{(m)}(A) = \mathrm{argmin}_{H_k^{(m)}} L_{k,m}^{(tr)}(A, \widetilde{W}_k^{(m)}(A), H_k^{(m)}, D_k^{(\mathrm{tr})}). \tag{5}$$

### 3.1.3 The final stage

At the final stage, each task $k$ evaluates the loss $L_k^{(val)}$ of its model learned in the final round $M$ (including encoder $\widetilde{W}_k^{(M)}(A)$, head $\widetilde{H}_k^{(M)}(A)$, architecture $A$) on its validation set $D_k^{(\mathrm{val})}$. $A$ is learned by minimizing the validation losses of all tasks:

$$\min_A \sum_{k=1}^K L_k^{(val)}(A, \widetilde{W}_k^{(M)}(A), \widetilde{H}_k^{(M)}(A), D_k^{(\mathrm{val})}). \tag{6}$$

In experiments, these losses for different tasks are normalized to have similar scales. Note that our method focuses on alleviating the burden of tuning the weights of tasks' training losses. As for the validation losses, we follow the standard practice of the multi-task learning literature, which treats each task's validation performance (after normalization) equally.

### 3.1.4 Multi-level optimization framework

To this end, we are ready to formulate the IMTNAS problem as a multi-level optimization framework. From bottom to top, the $K$ tasks perform $M$ rounds of interleaving learning. After the interleaving process is finished, the shared architecture $A$ is optimized by minimizing validation losses.

$$
\begin{aligned}
&\min_A \ \sum_{k=1}^K L_k^{(val)}(A, \widetilde{W}_k^{(M)}(A), \widetilde{H}_k^{(M)}(A), D_k^{(\text{val})}) \\
&s.t. \\
&\quad \cdots \\
&\quad \widetilde{H}_k^{(m)}(A) = \operatorname{argmin}_{H_k^{(m)}} L_{k,m}^{(tr)}(A, \widetilde{W}_k^{(m)}(A), H_k^{(m)}, D_k^{(\text{tr})}) \\
&\quad \widetilde{W}_k^{(m)}(A) = \operatorname{argmin}_{W_k^{(m)}} \sum_{\mathcal{A}_i, \mathcal{A}_j} \max\left(0, -\big(d(\mathcal{A}_i, \mathcal{A}_j; \widetilde{W}_{k-1}^{(m)}(A)) - \tau\big)\big(d(\mathcal{A}_i, \mathcal{A}_j; \widetilde{W}_k^{(m)}) - \tau\big)\right) \\
&\quad \cdots \\
&\quad \widetilde{W}_1^{(1)}(A), \widetilde{H}_1^{(1)}(A) = \operatorname{argmin}_{W_1^{(1)}, H_1^{(1)}} L_{1,1}^{(tr)}(A, W_1^{(1)}, H_1^{(1)}, D_1^{(\text{tr})})
\end{aligned}
\tag{7}
$$

The optimization algorithm for solving the problem in Eq.(7) is deferred to the appendix.

**Differentiable NAS.** In differentiable NAS (Liu et al., 2019a), the search space consists of many candidate building blocks, whose outputs are multiplied with architecture variables. The search process learns these variables using gradient methods. After learning, architecture blocks with largest variable values are retained to form the final architecture.

**Reduce computation and memory costs.** To reduce computation and memory costs, we adopt a hypernetwork (Ha et al., 2016) based approach. A hypernetwork (with weight parameters $V$ and a fixed architecture) takes a task number $k$, a round number $r$, and an architecture $A$ as inputs, and generates the weight parameters $W_k^{(m)}$ for $A$. In this way, we do not need to store $W_k^{(m)}$ (for $m = 1, \cdots, M$ and $k = 1, \cdots, K$) in memory, but only need to store $V$, which is much smaller. The hypernetwork is designed as follows. We put all weight parameters into a long vector (with dimension $D$), then partition it into $D/100$ sub-vectors, each with a dimension of 100. The hypernetwork is a feedforward network which takes the architecture $A$, task index $k$, round index $r$, sub-vector index $d$ as inputs and produces a 100-dimensional sub-vector. All the produced sub-vectors are concatenated to form the weight parameter vector. The hypernetwork has two hidden layers, with 30 and 50 hidden units respectively. The architecture $A$ is represented as a continuous vector. The dimension of the vector is the number of operations in the search space. $A_i$ denotes the importance weight of the $i$-th operation (i.e., whether it should be selected).

## 4 Experiments

In this section, we report experimental results. We experimented with two multi-task settings: 1) two heterogeneous tasks: image classification and object detection; 2) two homogeneous tasks: image classification on two different datasets. In Appendix B, we conduct experiments on three tasks: classification on ImageNet, CIFAR-100, and CIFAR-10. In all experiments, in Eq.(3), the number $C$ of augmentations in each set is set to 20. $\tau$ is set to 5. The kernel in MMD is set to a radial basis function kernel with the scale parameter set to 0.1. We use random crop (He et al., 2019), flip, rotation, and color jitter to generate augmentations for images. Please refer to the appendix for detailed hyperparameter settings and additional results, such as ablation study on the number $C$ of augmented examples, memory costs, increasing the number of rounds, how the MMD threshold $\tau$ affects experimental results, etc.

### 4.1 Datasets

For the image classification task, we used the following datasets: CIFAR-10 (Krizhevsky & Hinton, 2010), CIFAR-100 (Krizhevsky et al., 2009), and ImageNet (Deng et al., 2009). For the object detection task, we used the COCO (Lin et al., 2014) dataset. CIFAR-10 contains 60K images from 10 classes. CIFAR-100 contains 60K images from 100 classes. For CIFAR-10 and CIFAR-100, each of them is split into train/validation/test sets with 25K/25K/10K images respectively. ImageNet has 1.2M training images and 50K test images, from 1000 classes. COCO is broadly used for common object detection. It contains a training set (train2017) of 118K images and a test set (val2017) of 5K images, from 80 classes. Following many previous NAS methods (e.g., DARTS, PDARTS, PCDARTS) which split the training data of CIFAR-10 and CIFAR-100 into a new training set and a validation set with a ratio of 1:1, we randomly split the

118K COCO training set into a new training and validation set with a ratio of 1:1. Baseline methods (when applicable) also use the same split.

## 4.2 Baselines

We compare with the following baselines. 1) Multi-task learning (MTL) with soft weight sharing (MTL-SWS) (Maninis et al., 2019): tasks are encouraged to have similar encoder weights via L2 regularization. 2) MTL with hard weight sharing (MTL-HWS) (Kokkinos, 2017): different tasks share the same encoder weights. 3) MTL sharing architecture only (MTL-SAO): encoder weights of tasks are independent. 4) MTL-NAS (Gao et al., 2020): task-agnostic NAS for general-purpose MTL. 5) BMTAS (Bruggemann et al., 2020): branched multi-task architecture search. 6) Minibatch interleaving (MI) (Doire & Okubadejo, 2019): the model is the same as MTL-SWS; in the training process, parameters of different tasks are updated using their minibatches alternatively. 7) Separate interleaving (SI) (Tian et al., 2021): we perform task 1 first, then use the architecture and weights of its encoder to initialize that of task 2, which are then finetuned; this process applies to other tasks as well and interleaves for multiple rounds. 8) Performing two tasks independently: SingPathNAS (Stamoulis et al., 2019), MobileNetV3 (Howard et al., 2019), MnasNet-A2 (Tan et al., 2019b), MixNet-M (Tan & Le, 2019), and FairNAS (Chu et al., 2019a).

For MTL with soft weight sharing (MTL-SWS), the formulation is:

$$\min_A \ \sum_{k=1}^{K} \alpha_k L_k(A, \widetilde{W}_k(A), \widetilde{H}_k(A), D_k^{(\text{val})})$$
$$s.t. \ \{\widetilde{W}_k(A), \widetilde{H}_k(A)\}_{k=1}^{K} = \min_{\{W_k, H_k\}_{k=1}^{K}} \ \sum_{k=1}^{K} \beta_k L_k(A, W_k, H_k, D_k^{(\text{tr})}) + \gamma \sum_{1 \le k < j \le K} \|W_k - W_j\|_2^2$$

where $\{\alpha\}_{k=1}^K$, $\{\beta\}_{k=1}^K$, and $\gamma$ are tradeoff parameters. For MTL with hard weight sharing (MTL-HWS), the formulation is:

$$\min_A \ \sum_{k=1}^{K} \alpha_k L(A, \widetilde{W}(A), \widetilde{H}_k(A), D_k^{(\text{val})})$$
$$s.t. \ \widetilde{W}(A), \{\widetilde{H}_k(A)\}_{k=1}^{K} = \min_{W, \{H_k\}_{k=1}^{K}} \ \sum_{k=1}^{K} \beta_k L(A, W, H_k, D_k^{(\text{tr})}) \tag{8}$$

where $\{\alpha_k\}_{k=1}^K$ and $\{\beta_k\}_{k=1}^K$ are tradeoff parameters. For MTL sharing architecture only (MTL-SAO), the formulation is:

$$\min_A \ \sum_{k=1}^{K} \alpha_k L(A, \widetilde{W}_k(A), \widetilde{H}_k(A), D_k^{(\text{val})})$$
$$s.t. \ \{\widetilde{W}_k(A), \widetilde{H}_k(A)\}_{k=1}^{K} = \min_{\{W_k, H_k\}_{k=1}^{K}} \ \sum_{k=1}^{K} \beta_k L(A, W_k, H_k, D_k^{(\text{tr})}) \tag{9}$$

where $\{\alpha_k\}_{k=1}^K$ and $\{\beta_k\}_{k=1}^K$ are tradeoff parameters.

## 4.3 MTL on image classification and object detection

In this section, the two tasks that are interleaved over are: image classification on ImageNet and object detection on COCO.

### 4.3.1 Experimental settings

For encoder architecture, its search space is the same as that in ProxylessNAS (Cai et al., 2019), which contains 19 layers and each layer contains 7 choices. For objection detection head (ODH), we use the one in RetinaNet (Lin et al., 2017). Hyperparameter settings for ODH are the same as those in (Lin et al., 2017). The classification head for ImageNet is a 1000-way linear classifier. We use 3 interleaving rounds, with an order of "cls,det,cls,···", where "cls" and "det" denote classification and detection tasks respectively. For image classification, we use top-1 and top-5 errors for evaluation. For object detection, we use Average Precision (AP) (Ren et al., 2015) as an evaluation metric, across IoU thresholds from 0.5 to 0.95 with an interval of 0.05 and different scales (small, medium, large). The model was trained for 40 epochs. Network weights were trained using SGD, with an initial learning rate of 0.02 (with cosine decay). Architecture variables were trained using Adam (Kingma & Ba, 2014) with a learning rate of 5e-4. Batch size was 1024 for ImageNet and 12 for COCO. In MTL-SWS, tasks 1 and 2 are classification and detection. $\alpha_1$, $\alpha_2$,

Table 2: Object detection on COCO test set and classification accuracy on ImageNet test set. APS, APM, APL: AP at small, medium, large scale. AP50 and AP75 are AP with IoU thresholds of 0.5 and 0.75. Acc is accuracy.

| | COCO | | | | | | | ImageNet | | ×+ |
| --- | --- | --- | --- | --- | --- | --- | --- | --- | --- | --- |
| | Acc | AP | AP50 | AP75 | APS | APM | APL | Top-1 | Top-5 | (M) |
| SingPathNAS (Stamoulis et al., 2019) | 75.0 | 30.7 | 49.8 | 32.2 | 15.4 | 33.9 | 41.6 | 75.0 | 92.2 | 365 |
| MobileNetV3 (Howard et al., 2019) | 75.2 | 29.9 | 49.3 | 30.8 | 14.9 | 33.3 | 41.1 | 75.2 | 92.2 | 219 |
| MnasNet-A2 (Tan et al., 2019b) | 75.6 | 30.5 | 50.2 | 32.0 | 16.6 | 34.1 | 41.1 | 75.6 | 92.7 | 340 |
| MixNet-M (Tan & Le, 2019) | 77.0 | 31.3 | 51.7 | 32.4 | 17.0 | 35.0 | 41.9 | 77.0 | 93.3 | 360 |
| FairNAS-C (Chu et al., 2019a) | 76.7 | 31.2 | 50.8 | 32.7 | 16.3 | 34.4 | 42.3 | 76.7 | 93.3 | 325 |
| MTL-SAO | 76.8 | 31.0 | 51.2 | 33.4 | 16.1 | 34.6 | 41.6 | 76.3 | 92.8 | 379 |
| MTL-HWS (Kokkinos, 2017) | 77.0 | 31.5 | 51.5 | 33.2 | 16.5 | 34.8 | 41.7 | 76.8 | 93.1 | 384 |
| MTL-SWS (Maninis et al., 2019) | 77.3 | 31.9 | 51.8 | 33.8 | 16.6 | 35.1 | 42.1 | 77.1 | 93.4 | 395 |
| MTL-NAS (Gao et al., 2020) | 77.2 | 31.3 | 51.1 | 33.5 | 16.1 | 34.4 | 41.7 | 76.5 | 93.0 | 495 |
| BMTAS (Bruggemann et al., 2020) | 76.3 | 30.5 | 51.0 | 33.1 | 16.2 | 34.3 | 41.3 | 76.3 | 92.6 | 432 |
| MI (Doire & Okubadejo, 2019) | 77.1 | 31.4 | 51.5 | 33.4 | 16.4 | 34.7 | 41.6 | 76.8 | 93.2 | 389 |
| SI (Tian et al., 2021) | 77.1 | 31.1 | 50.6 | 32.7 | 15.8 | 34.1 | 41.4 | 76.5 | 92.7 | 371 |
| IMTNAS (ours) | **77.7** | **32.5** | **53.1** | **34.1** | **17.1** | **36.8** | **43.4** | **77.8** | **93.9** | 381 |

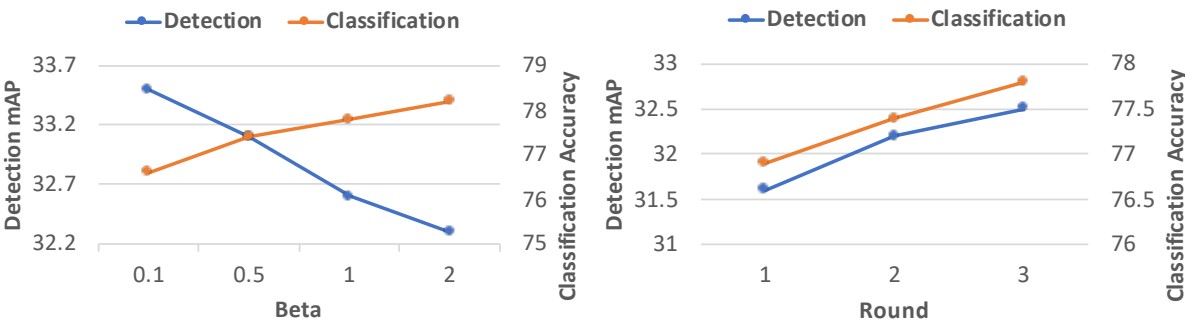

Figure 1: (Left) In MTL-SWS, improving one task leads to performance degradation of the other task. (Right) In our method, how task performance varies with the number of rounds.

$\beta_1$, $\beta_2$, and $\gamma$ are set to 1, 0.5, 1, 0.5, and 10 respectively. For non-MTL baselines, we mostly follow their original hyperparameter settings.

### 4.3.2 Results and analysis

Table 2 shows object detection results on the COCO test set and classification results on the ImageNet test set. From this table, we make the following observations. **First**, IMTNAS performs better than MTL-SWS. In MTL-SWS, object detection and image classification are performed at the same time by minimizing the weighted sum of their losses. More loss decrease of one task leads to less loss decrease of the other task. To verify this, we plot how test performances of these two tasks (mAP of detection task on left axis and accuracy of classification task on right axis) vary with the weight $\beta$ of classification task loss (weight of detection loss is set to 1) in Figure 1(Left). As $\beta$ increases (more attention is paid to the classification task), top-1 test accuracy on ImageNet increases while test mAP on COCO decreases. This demonstrates that improving the performance of one task leads to the performance degradation of the other task. In contrast, each loss in our method focuses on learning one task at a time, thus can avoid this problem.

On the other hand, the interleaving mechanism in IMTNAS enables each task to transfer its knowledge to any other task along the interleaving loop. Figure 1(Right) shows the test performance of models in a 3-round interleaving process. As the number of rounds increases, detection models and classification models perform better. This demonstrates that via interleaving, each task continuously improves by leveraging knowledge from previous tasks. From Figure 1(Right), we see that our method avoids catastrophic forgetting (Kirkpatrick et al., 2017). This is because in our method, all

Table 3: Classification errors (%) on test sets of CIFAR-100 and CIFAR-10, number of model parameters (millions), and search cost (GPU days on a Nvidia 1080Ti, sum of costs of all tasks). Results on ImageNet are under mobile settings. IMTNAS-Darts2nd: our IMTNAS is applied to DARTS-2nd. Results marked with * are taken from DARTS$^-$ (Chu et al., 2020a), DrNAS (Chen et al., 2020b), $\beta$-DARTS (Ye et al., 2022), and AGNAS (Sun et al., 2022). Methods marked with † were re-run 10 times with random initialization.

| | CIFAR100 Error(%) | CIFAR10 Error(%) | ImageNet Top-1 | Top-5 | Param (M) | Cost (GPU days) |
|---|---|---|---|---|---|---|
| *R-DARTS (Zela et al., 2020) | 17.98±0.21 | 2.92±0.24 | - | - | - | 3.3 |
| *DARTS+PT (Wang et al., 2021) | - | 2.61±0.08 | - | - | 3.0 | 0.8 |
| *ISTA-NAS (Yang et al., 2020) | - | 2.51±0.04 | 23.9 | 7.1 | 3.3 | 3.3 |
| *MiLeNAS (He et al., 2020) | - | 2.49±0.08 | 24.7 | 7.6 | 3.9 | 3.3 |
| *GAEA (Li et al., 2021) | - | 2.49±0.04 | 24.0 | 7.2 | - | 3.3 |
| *PDARTS-ADV (Chen & Hsieh, 2020) | - | 2.44±0.05 | 24.2 | 7.2 | 3.4 | 3.3 |
| *DOTS (Gu et al., 2021) | 16.48±0.13 | 2.49±0.06 | 24.3 | 7.4 | 4.1 | 0.3 |
| *$\beta$-DARTS (Ye et al., 2022) | 16.24±0.22 | 2.53±0.08 | 23.9 | 7.0 | 3.8 | 0.4 |
| *AGNAS (Sun et al., 2022) | - | 2.53±.003 | 23.4 | 6.8 | 3.6 | 0.4 |
| MTL-NAS (Gao et al., 2020) | 19.44±0.30 | 2.89±0.09 | 26.6 | 8.8 | 4.8 | 3.4 |
| BMTAS (Bruggemann et al., 2020) | 20.12±0.21 | 2.84±0.15 | 27.0 | 9.0 | 4.4 | 3.3 |
| *Darts2nd (Liu et al., 2019a) | 20.54±0.41 | 2.75±0.07 | 26.7 | 8.6 | 3.3 | 8.2 |
| MTL-SAO-Darts2nd | 18.87±0.21 | 2.88±0.10 | 26.4 | 8.5 | 3.4 | 8.3 |
| MTL-HWS-Darts2nd (Kokkinos, 2017) | 18.78±0.30 | 2.86±0.06 | 26.9 | 8.8 | 3.3 | 8.1 |
| MTL-SWS-Darts2nd (Maninis et al., 2019) | 18.74±0.09 | 2.73±0.11 | 26.2 | 8.3 | 3.2 | 8.4 |
| MI-Darts2nd (Doire & Okubadejo, 2019) | 18.83±0.26 | 2.87±0.14 | 26.8 | 8.8 | 3.5 | 8.4 |
| SI-Darts2nd (Tian et al., 2021) | 19.16±0.26 | 2.94±0.05 | 26.9 | 8.9 | 3.1 | 8.9 |
| IMTNAS-Darts2nd (ours) | **17.33**±0.10 | **2.65**±0.02 | **25.6** | **8.0** | 3.2 | 8.2 |
| †Pdarts (Chen et al., 2019) | 17.48±0.18 | 2.53±0.06 | 24.4 | 7.3 | 3.6 | 0.7 |
| MTL-SAO-Pdarts | 17.65±0.29 | 2.61±0.06 | 24.9 | 7.9 | 3.5 | 0.7 |
| MTL-HWS-Pdarts (Kokkinos, 2017) | 17.54±0.10 | 2.66±0.09 | 25.3 | 8.0 | 3.7 | 0.7 |
| MTL-SWS-Pdarts (Maninis et al., 2019) | 17.49±0.12 | 2.60±0.04 | 24.5 | 7.4 | 3.5 | 0.7 |
| MI-Pdarts (Doire & Okubadejo, 2019) | 17.51±0.11 | 2.67±0.12 | 25.4 | 8.2 | 3.7 | 0.7 |
| SI-Pdarts (Tian et al., 2021) | 17.79±0.41 | 2.69±0.10 | 25.8 | 8.5 | 3.6 | 0.9 |
| IMTNAS-Pdarts (ours) | **16.30**±0.09 | **2.45**±0.03 | **23.6** | **6.8** | 3.5 | 0.7 |
| †Pcdarts (Xu et al., 2020) | 17.92±0.13 | 2.51±0.05 | 24.3 | 7.4 | 3.9 | 0.6 |
| MTL-SAO-Pcdarts | 18.06±0.22 | 2.64±0.10 | 24.6 | 7.6 | 3.8 | 0.6 |
| MTL-HWS-Pcdarts (Kokkinos, 2017) | 18.14±0.15 | 2.69±0.12 | 24.7 | 7.8 | 4.0 | 0.6 |
| MTL-SWS-Pcdarts (Maninis et al., 2019) | 18.00±0.41 | 2.51±0.11 | 24.3 | 7.3 | 3.8 | 0.6 |
| MI-Pcdarts (Doire & Okubadejo, 2019) | 18.19±0.23 | 2.70±0.09 | 24.8 | 7.8 | 4.0 | 0.6 |
| SI-Pcdarts (Tian et al., 2021) | 17.94±0.11 | 2.67±0.05 | 25.1 | 8.1 | 4.1 | 0.7 |
| IMTNAS-Pcdarts (ours) | **17.31**±0.08 | **2.43**±0.03 | **23.5** | **6.6** | 3.9 | 0.6 |
| †Prdarts (Zhou et al., 2020) | 16.41±0.08 | 2.36±0.03 | 24.1 | 7.3 | 3.4 | 0.5 |
| MTL-SAO-Prdarts | 17.05±0.11 | 2.64±0.09 | 24.5 | 7.8 | 3.6 | 0.5 |
| MTL-HWS-Prdarts (Kokkinos, 2017) | 16.93±0.12 | 2.55±0.06 | 24.3 | 7.4 | 3.7 | 0.5 |
| MTL-SWS-Prdarts (Maninis et al., 2019) | 16.71±0.13 | 2.50±0.09 | 24.3 | 7.3 | 3.4 | 0.5 |
| MI-Prdarts (Doire & Okubadejo, 2019) | 17.32±0.17 | 2.76±0.14 | 26.1 | 8.1 | 3.5 | 0.6 |
| SI-Prdarts (Tian et al., 2021) | 17.25±0.11 | 2.70±0.07 | 25.8 | 8.0 | 3.5 | 0.8 |
| IMTNAS-Prdarts (ours) | **15.97**±0.06 | **2.32**±0.03 | **23.4** | **6.3** | 3.3 | 0.5 |

tasks are learned jointly in a unified framework where every task can transfer knowledge to any other task and all tasks learn the shared encoder architecture together.

**Second**, our method performs better than MTL-HWS. It is because MTL-HWS requires two tasks to share exactly the same encoder, which is not good for task-specific representation learning. The datasets of different tasks have different properties. We need to use task-specific encoders with different weight parameters to learn task-specific representations for these datasets. MTL-HWS, which makes the encoders of all tasks share the same weights, compromises this purpose. **Third**, our method outperforms MTL-SAO. In MTL-SAO, two tasks only share encoder architecture and

do not share commonality in encoder weights. Weights also carry important knowledge, which should be transferred between tasks. **Fourth**, our method performs better than MTL-NAS and BMTAS. Similar to MTL-SWS, in these two methods, more decrease of loss in one task renders less decrease of loss in the other task. **Fifth**, our method achieves better performance than SI. In SI, different tasks are trained separately while our method performs two tasks in an end-to-end framework. **Sixth**, our method outperforms baselines that perform two tasks independently. These methods do not conduct knowledge transfer between tasks. **Seventh**, the computational demand (the number of composite multiply-accumulate ($\times+$) operations for a single image) of our method is comparable with baselines.

### 4.4 Multi-task classification on CIFAR-100 and CIFAR-10

In this section, the two tasks that are interleaved over are: image classification on CIFAR-100 and image classification on CIFAR-10.

#### 4.4.1 Experimental settings

Following the experimental protocol in (Liu et al., 2019a), each experiment consists of an architecture search phase and an architecture evaluation phase. In the search phase, an optimal architecture cell is searched by minimizing a validation loss. In the evaluation phase, a larger network is created by stacking multiple copies of an optimally searched cell. This new network is re-trained from scratch and evaluated on a test set. We experimented with the search spaces in DARTS (Liu et al., 2019a), P-DARTS (Chen et al., 2019), and PC-DARTS (Xu et al., 2020). The classification head for CIFAR-10 and CIFAR-100 is a 10-way and 100-way linear classifier respectively. We set the number of interleaving rounds to 2, with the following task order: C10,C100,C10,C100. During architecture search, network weights were optimized using the SGD optimizer with a batch size of 64, an initial learning rate of 0.025, a learning rate scheduler of cosine decay, a weight decay of 3e-4, a momentum of 0.9, and an epoch number of 50 (only for our method; the number of epochs for baselines are given in the appendix). Architecture variables were optimized using the Adam (Kingma & Ba, 2014) optimizer with a learning rate of 3e-4 and a weight decay of 1e-3. The rest of hyperparameters follow those in DARTS, P-DARTS, and PC-DARTS. The experiments were conducted on one Nvidia 1080Ti GPU. Mean and standard deviation of classification errors obtained from 10 random runs are reported. For baselines, we ran them longer than the time originally reported in their papers to ensure a fair comparison. In MTL-SWS, tasks 1 and 2 are classification on CIFAR-100 and CIFAR-10. $\alpha_1$, $\alpha_2$, $\beta_1$, $\beta_2$, and $\gamma$ are set to 1, 1, 1, 1, and 100 respectively.

Table 4: Test errors of MTL-SWS-Darts2nd (%). The weight of Cifar100 (C100) is set to 1. The weight $\beta_2$ of Cifar10 (C10) is tuned. Error(Overall)=Error(C100)+6*Error(C10).

| $\beta_2$ | 0.01 | 0.05 | 0.1 | 0.5 | 0.7 | 1 | 1.5 | 2 |
|---|---|---|---|---|---|---|---|---|
| C100 Error | 17.5 | 17.7 | 18.0 | 18.1 | 18.7 | 18.8 | 19.6 | 20.1 |
| C10 Error | 3.21 | 3.06 | 2.99 | 2.93 | 2.87 | 2.75 | 2.73 | 2.70 |
| Overall Error | 36.8 | 36.1 | 35.9 | 35.7 | 35.9 | **35.3** | 36.0 | 36.3 |

#### 4.4.2 Results and analysis

Table 3 shows results on CIFAR-100 and CIFAR-10. When our proposed IMTNAS framework is applied to different differentiable NAS methods, the errors of these methods can be greatly reduced, which demonstrates the effectiveness of IMTNAS. In the interleaving process, each task transfers its knowledge to the next task. To ensure successful transfer, each task needs to learn representations not only good for itself, but also good for the next task. By doing this, the learned representations have good generalizability across tasks.

In MTL-SWS-Darts2nd, we tuned task weights. Table 4 shows that the weight needs

Table 5: Comparison between hypernetwork (HN) based approaches and non-hypernetwork (NHN) approaches.

| | Param. (M) | Search Cost (GPU days) | Memory (MiB) |
|---|---|---|---|
| Darts2nd (Liu et al., 2019a) | 3.3 | 8.2 | 22071 |
| NHN-Darts2nd (ours) | 3.3 | 13.9 | 39418 |
| HN-Darts2nd (ours) | 3.2 | 8.2 | 22129 |
| Pdarts (Chen et al., 2019) | 3.6 | 0.7 | 19335 |
| NHN-Pdarts (ours) | 3.5 | 1.2 | 37529 |
| HN-Pdarts (ours) | 3.5 | 0.7 | 19391 |
| Pcdarts (Xu et al., 2020) | 3.9 | 0.6 | 20636 |
| NHN-Pcdarts (ours) | 4.1 | 1.1 | 39773 |
| HN-Pcdarts (ours) | 3.9 | 0.6 | 20701 |
| Prdarts (Zhou et al., 2020) | 3.4 | 0.5 | 20658 |
| NHN-Prdarts (ours) | 3.5 | 0.9 | 39271 |
| HN-Prdarts (ours) | 3.3 | 0.5 | 20674 |

to be tuned extensively for achieving optimal overall performance. In contrast, our method does not need to tune weights of tasks.

Other observations made from Table 3 are similar to those from Table 2: 1) our method works better than MTL baselines including MTL-SAO, MTL-HWS, MTL-SWS, SI, MTL-NAS, and BMTAS; 2) our method outperforms baselines which conduct different tasks separately. The analysis of reasons is

Table 6: Ablation on knowledge transfer approaches.

| Method | CIFAR-100 Error (%) | CIFAR-10 Error (%) |
|---|---|---|
| Ours | **17.33**±0.10 | **2.65**±0.02 |
| L2-Weights | 18.46±0.09 | 2.72±0.05 |
| L2-Embedding | 18.31±0.13 | 2.73±0.05 |
| Example-Similarity | 18.19±0.10 | 2.71±0.06 |

similar to that for results in Table 2. The computational costs and number of model parameters of our methods are similar to those of baselines, while our method achieves better accuracy.

Following (Liu et al., 2019a), we assess transferability of architectures searched on CIFAR100/10 by evaluating them on ImageNet. From ImageNet results in Table 3, we can see that architectures searched by our methods are highly transferable, which achieve lower errors than 1) vanilla DARTS, P-DARTS, PC-DARTS; 2) MTL baselines; and 3) non-MTL baselines.

### 4.4.3 Ablation studies

We compare the search and memory costs of our hypernetwork (HN) based approaches with those of non-hypernetwork (NHN) approaches that store $W_k^{(m)}$ in memory. Table 5 shows the results. As can be seen, hypernetwork-based approaches have much lower search and

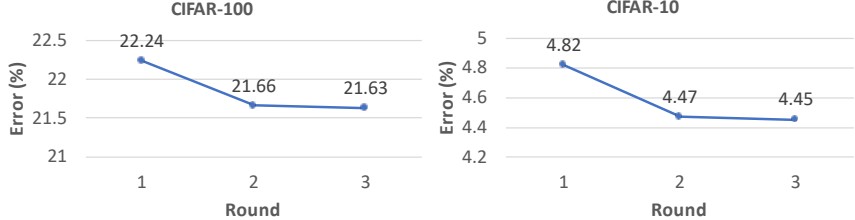

Figure 2: How classification errors (%) on CIFAR-100 and CIFAR-10 change with the number of interleaving rounds $M$.

memory costs than non-hypernetwork approaches while the test errors of hypernetwork-based approaches are similar to those of non-hypernetwork approaches as shown in Table 3.

We compare our distribution matching based knowledge transfer (KT) approach with several other KT methods, including 1) L2-Weights (Rajeswaran et al., 2019): encouraging the L2 distance between $\widetilde{W}_{k-1}^{(m)}(A)$ and $W_k^{(m)}$ to be small; 2) L2-Embedding (Romero et al., 2014): encouraging the embeddings of input images extracted by $\widetilde{W}_{k-1}^{(m)}(A)$ and $W_k^{(m)}$ to have small L2 distance; 3) Example-Similarity (Chen et al., 2020a): using $\widetilde{W}_{k-1}^{(m)}(A)$ to label whether a pair of input examples are similar and learn $W_k^{(m)}$ by fitting these labels. The experiments are conducted on DARTS. Table 6 shows the results. Our method outperforms L2-Weights and L2-Embedding because these two baselines impose strong restrictions that weights or embeddings need to be close in an L2 sense while our method is more flexible. Our method works better than Example-Similarity.
This method is limited to capturing low-order relationships between data examples (the order is two specifically, since pairwise similarity is measured). In contrast, our method measures similarity at the distribution level instead of individual example level, therefore can capture high-order relationships between data examples and more holistic properties of an entire data distribution.

Table 7: Ablation study on task order. "Order 1" denotes "C100, C10, C100, C10". "Order 2" denotes "C10, C100, C10, C100".

| Method | Error (%) |
|---|---|
| Order 1 (CIFAR-100) | 17.33±0.10 |
| Order 2 (CIFAR-100) | 17.21±0.18 |
| Order 1 (CIFAR-10) | 2.65±0.02 |
| Order 2 (CIFAR-10) | 2.67±0.12 |

We study how the test error of the final model in an interleaving sequence changes with the number of interleaving rounds $M$. Results are reported on 10K held-out data. IMTNAS is applied to DRATS-2nd. Figure 2 shows the results. When $M$ increases from 1 to 2, the errors on CIFAR-100/10 are reduced. When $M = 1$, the interleaving effect is weak: classification on CIFAR-100 influences classification on CIFAR-10, but not the other way around. When $M = 2$, the interleaving effect is strong: CIFAR-100 influences CIFAR-10 and CIFAR-10 in turn influences CIFAR-100. This further demonstrates the effectiveness of interleaving learning. Increasing $M$ from 2 to 3 does not significantly reduce the errors further. This is probably because 2 rounds of interleaving have brought in a sufficient interleaving effect.

Next, we explore whether task order affects performance. In IMTNAS-Darts2nd, we experimented with two orders (with the number of rounds set to 2): 1) CIFAR-100, CIFAR-10, CIFAR-100, CIFAR-10; 2) CIFAR-10, CIFAR-100, CIFAR-10, CIFAR-100. Table 7 shows the test errors under two different orders. As can be seen, the errors are

Table 8: Test errors on CIFAR-100 and CIFAR-10 under different values of $C$.

| C | 0 | 3 | 10 | 20 | 30 | 40 |
|---|---|---|---|---|---|---|
| C100 Error | 20.71 | 18.22 | 17.64 | **17.33** | 17.36 | 17.29 |
| C10 Error | 2.77 | 2.72 | 2.66 | **2.65** | 2.67 | 2.70 |

not affected by the task order significantly. The reason is: via interleaving, each task influences the other task at some point in the interleaving sequence; therefore, it does not matter too much regarding which task should be performed first.

We also investigate how the number $C$ of augmented examples during knowledge transfer affects model performance. The study is conducted on

Table 9: Test errors on Cifar100/10 under different round numbers.

| | Round number | Epoch number | Cifar100 error | Cifar10 error |
|---|---|---|---|---|
| Ablation study | 20 | 5 | 24.55±0.11 | 5.24±0.06 |
| Main study | 2 | 50 | **17.16**±0.07 | **2.63**±0.03 |

Ours+Darts which performs interleaving learning on CIFAR-100 and CIFAR-10. Table 8 shows a moderate $C$ yields the lowest overall errors. As can be seen, when $C$ is small, the performance is not very good. This is because under a small $C$, a smaller number of augmented examples cannot represent a distribution very well. When $C$ is more than 20, the performance does not improve significantly, but the computational cost will increase. So in our experiments, we set $C$ to 20. For $C = 0$, no knowledge is transferred, which leads to worse performance.

We perform a study which increases the number of interleaving rounds to 20. Meanwhile, to keep the computational cost of this ablation study the same as that of the study (referred to as main study) corresponding to Table 3 where the number of rounds is 2 and the epoch number is 50, we reduce the number of epochs in this ablation study to 5. This ablation study is performed on Ours+Darts, which performs interleaving learning on CIFAR-100 and CIFAR-10. Table 9 shows the results. As can be seen, increasing the number of rounds to 20 leads to worse performance. The reason is: 1) epoch number is too small; 2) it is not very necessary to perform 20 rounds of interleaving; two rounds are sufficient for enabling each task to transfer knowledge to any other task.

Finally, we investigate how the MMD threshold $\tau$ in Eq.(4) affects performance. The study is performed on Ours+Darts, which performs interleaving learning on CIFAR-100 and CIFAR-10. Table 10 shows the results. As can be seen, the value of $\tau$ does not have a significant influence on the performance. $\tau$ is like the margin in a support vector machine; its absolute value does not matter much. When $\tau$ increases or decreases, the scale

Table 10: Test errors on CIFAR-100 and CIFAR-10 under different values of $\tau$.

| $\tau$ | 0.1 | 1 | 5 | 25 | 100 |
|---|---|---|---|---|---|
| C100 Error | 17.36 | 17.35 | 17.33 | 17.33 | 17.35 |
| C10 Error | 2.67 | 2.66 | 2.65 | 2.65 | 2.67 |

of MMD can increase or decrease accordingly (by scaling the norm of latent representations $z_n^{(i)}$ and $z_m^{(j)}$ up or down accordingly) to match with the scale of $\tau$.

## 5 Conclusions and discussions

We propose an interleaving multi-task neural architecture search framework, where multiple tasks are performed in an interleaving fashion. Via interleaving, different models transfer their knowledge to each other. Experiments on various datasets demonstrate the efficacy of our method. One major limitation of this work is that it requires the NAS methods to be differentiable. For non-differentiable NAS methods such as those based on reinforcement learning and evolutionary algorithms, our method is not applicable.

**Broader Impact Statement**

One potential negative societal impact of our work is: our method currently does not take interpretability of searched architectures into account, which may yield less reliable predictions and incur trustworthiness concerns in mission-critical domains such as healthcare and finance.

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

# A  Optimization algorithm

We use a well-established algorithm developed in (Liu et al., 2019a) to solve the interleaving learning problem. Theoretic convergence of this algorithm has been broadly analyzed in (Ghadimi & Wang, 2018; Grazzi et al., 2020; Ji et al., 2021; Liu et al., 2021; Yang et al., 2021). For the ease of presentation, we first present the algorithm for a special case of our method where the number of tasks is two and the number of interleaving rounds is two as well. Extensions to more tasks and more rounds can be straightforwardly conducted. In the second subsection, we will introduce a general version of this algorithm. At each level of optimization problem, the optimal solution (on the left-hand side of the equal sign, marked with $\widetilde{\cdot}$), its exact value is computationally expensive to compute. To address this problem, following (Liu et al., 2019a), we approximate the optimal solution using a one-step gradient descent update and plug the approximation into the next level of optimization problem. In the sequel, $\frac{\partial \cdot}{\partial \cdot}$ denotes partial derivative. $\frac{d \cdot}{d \cdot}$ denotes an ordinary derivative.

### A.1 Special Case

We consider the following special case of our method with two tasks and two interleaving rounds.

$$\min_A L_1^{(val)}(A, \widetilde{W}_1^{(2)}(A), \widetilde{H}_1^{(2)}(A), D_1^{(\text{val})}) + L_2^{(val)}(A, \widetilde{W}_2^{(2)}(A), \widetilde{H}_2^{(2)}(A), D_2^{(\text{val})})$$
$$s.t.$$
$$\widetilde{H}_2^{(2)}(A) = \operatorname{argmin}_{H_2^{(2)}} L_{2,2}^{(tr)}(A, \widetilde{W}_2^{(2)}(A), H_2^{(2)}, D_2^{(\text{tr})})$$
$$\widetilde{W}_2^{(2)}(A) = \operatorname{argmin}_{W_2^{(2)}} \sum_{\mathcal{A}_i,\mathcal{A}_j} \max\left(0, -\left(d(\mathcal{A}_i, \mathcal{A}_j; A, \widetilde{W}_1^{(2)}(A)) - \tau\right)\left(d(\mathcal{A}_i, \mathcal{A}_j; A, W_2^{(2)}) - \tau\right)\right)$$
$$\widetilde{H}_1^{(2)}(A) = \operatorname{argmin}_{H_1^{(2)}} L_{1,2}^{(tr)}(A, \widetilde{W}_1^{(2)}(A), H_1^{(2)}, D_1^{(\text{tr})})$$
$$\widetilde{W}_1^{(2)}(A) = \operatorname{argmin}_{W_1^{(2)}} \sum_{\mathcal{A}_i,\mathcal{A}_j} \max\left(0, -\left(d(\mathcal{A}_i, \mathcal{A}_j; A, \widetilde{W}_2^{(1)}(A)) - \tau\right)\left(d(\mathcal{A}_i, \mathcal{A}_j; A, W_1^{(2)}) - \tau\right)\right) \quad (10)$$
$$\widetilde{H}_2^{(1)}(A) = \operatorname{argmin}_{H_2^{(1)}} L_{2,1}^{(tr)}(A, \widetilde{W}_2^{(1)}(A), H_2^{(1)}, D_2^{(\text{tr})})$$
$$\widetilde{W}_2^{(1)}(A) = \operatorname{argmin}_{W_2^{(1)}} \sum_{\mathcal{A}_i,\mathcal{A}_j} \max\left(0, -\left(d(\mathcal{A}_i, \mathcal{A}_j; A, \widetilde{W}_1^{(1)}(A)) - \tau\right)\left(d(\mathcal{A}_i, \mathcal{A}_j; A, W_2^{(1)}) - \tau\right)\right)$$
$$\widetilde{W}_1^{(1)}(A), \widetilde{H}_1^{(1)}(A) = \operatorname{argmin}_{W_1^{(1)}, H_1^{(1)}} L_{1,1}^{(tr)}(A, W_1^{(1)}, H_1^{(1)}, D_1^{(\text{tr})})$$

Next, we present the algorithm for solving this special case. We approximate $\widetilde{W}_1^{(1)}(A)$ using one-step gradient descent update of $W_1^{(1)}$ w.r.t $L_{1,1}^{(tr)}(A, W_1^{(1)}, H_1^{(1)}, D_1^{(\text{tr})})$:

$$\widetilde{W}_1^{(1)}(A) \approx \widehat{W}_1^{(1)} = W_1^{(1)} - \eta_w \nabla_{W_1^{(1)}} L_{1,1}^{(tr)}(A, W_1^{(1)}, H_1^{(1)}, D_1^{(\text{tr})}) \quad (11)$$

We update $H_1^{(1)}$ using gradient descent:

$$H_1^{(1)} \leftarrow H_1^{(1)} - \eta_h \nabla_{H_1^{(1)}} L_{1,1}^{(tr)}(A, W_1^{(1)}, H_1^{(1)}, D_1^{(\text{tr})}) \quad (12)$$

We plug the approximation $\widetilde{W}_1^{(1)}(A) \approx \widehat{W}_1^{(1)}$ into $\sum_{\mathcal{A}_i,\mathcal{A}_j} \max\left(0, -\left(d(\mathcal{A}_i, \mathcal{A}_j; A, \widetilde{W}_1^{(1)}(A)) - \tau\right)\left(d(\mathcal{A}_i, \mathcal{A}_j; A, W_2^{(1)}) - \tau\right)\right)$ and get an approximated objective. Then we approximate $\widetilde{W}_2^{(1)}(A)$ using one-step gradient descent update of $W_2^{(1)}$:

$$\widetilde{W}_2^{(1)}(A) \approx \widehat{W}_2^{(1)} = W_2^{(1)} - \eta_w \nabla_{W_2^{(1)}} \sum_{\mathcal{A}_i,\mathcal{A}_j} \max\left(0, -\left(d(\mathcal{A}_i, \mathcal{A}_j; A, \widehat{W}_1^{(1)}) - \tau\right)\left(d(\mathcal{A}_i, \mathcal{A}_j; A, W_2^{(1)}) - \tau\right)\right) \quad (13)$$

We plug the approximation $\widetilde{W}_2^{(1)}(A) \approx \widehat{W}_2^{(1)}$ into $L_{2,1}^{(tr)}(A, \widetilde{W}_2^{(1)}(A), H_2^{(1)}, D_2^{(\text{tr})})$ and get an approximated objective. Then we approximate $\widetilde{H}_2^{(1)}(A)$ using one-step gradient descent update of $H_2^{(1)}$:

$$\widetilde{H}_2^{(1)}(A) \approx \widehat{H}_2^{(1)} = H_2^{(1)} - \eta_h \nabla_{H_2^{(1)}} L_{2,1}^{(tr)}(A, \widehat{W}_2^{(1)}, H_2^{(1)}, D_2^{(\text{tr})}) \quad (14)$$

We plug the approximation $\widetilde{W}_2^{(1)}(A) \approx \widehat{W}_2^{(1)}$ into $\sum_{\mathcal{A}_i,\mathcal{A}_j} \max\left(0, -\left(d(\mathcal{A}_i, \mathcal{A}_j; A, \widetilde{W}_2^{(1)}(A)) - \tau\right)\left(d(\mathcal{A}_i, \mathcal{A}_j; A, W_1^{(2)}) - \tau\right)\right)$ and get an approximated objective. Then we approximate $\widetilde{W}_1^{(2)}(A)$ using one-step gradient descent update of $W_1^{(2)}$:

$$\widetilde{W}_1^{(2)}(A) \approx \widehat{W}_1^{(2)} = W_1^{(2)} - \eta_w \sum_{\mathcal{A}_i,\mathcal{A}_j} \max\left(0, -\left(d(\mathcal{A}_i, \mathcal{A}_j; A, \widehat{W}_2^{(1)}) - \tau\right)\left(d(\mathcal{A}_i, \mathcal{A}_j; A, W_1^{(2)}) - \tau\right)\right) \quad (15)$$

We plug the approximation $\widetilde{W}_1^{(2)}(A) \approx \widehat{W}_1^{(2)}$ into $L_{1,2}^{(tr)}(A, \widetilde{W}_1^{(2)}(A), H_1^{(2)}, D_1^{(\text{tr})})$ and get an approximated objective. Then we approximate $\widetilde{H}_1^{(2)}(A)$ using one-step gradient descent update of $H_1^{(2)}$:

$$\widetilde{H}_1^{(2)}(A) \approx \widehat{H}_1^{(2)} = H_1^{(2)} - \eta_h \nabla_{H_1^{(2)}} L_{1,2}^{(tr)}(A, \widehat{W}_1^{(2)}, H_1^{(2)}, D_1^{(\text{tr})}) \quad (16)$$

We plug the approximation $\widetilde{W}_1^{(2)}(A) \approx \widehat{W}_1^{(2)}$ into $\sum_{\mathcal{A}_i, \mathcal{A}_j} \max\left(0, -\left(d(\mathcal{A}_i, \mathcal{A}_j; A, \widetilde{W}_1^{(2)}(A)) - \tau\right)\left(d(\mathcal{A}_i, \mathcal{A}_j; A, W_2^{(2)}) - \tau\right)\right)$ and get an approximated objective. Then we approximate $\widetilde{W}_2^{(2)}(A)$ using one-step gradient descent update of $W_2^{(2)}$:

$$\widetilde{W}_2^{(2)}(A) \approx \widehat{W}_2^{(2)} = W_2^{(2)} - \eta_w \nabla_{W_2^{(2)}} \sum_{\mathcal{A}_i, \mathcal{A}_j} \max\left(0, -\left(d(\mathcal{A}_i, \mathcal{A}_j; A, \widehat{W}_1^{(2)}) - \tau\right)\left(d(\mathcal{A}_i, \mathcal{A}_j; A, W_2^{(2)}) - \tau\right)\right) \quad (17)$$

We plug the approximation $\widetilde{W}_2^{(2)}(A) \approx \widehat{W}_2^{(2)}$ into $L_{2,2}^{(tr)}(A, \widetilde{W}_2^{(2)}(A), H_2^{(2)}, D_2^{(\mathrm{tr})})$ and get an approximated objective. Then we approximate $\widetilde{H}_2^{(2)}(A)$ using one-step gradient descent update of $H_2^{(2)}$:

$$\widetilde{H}_2^{(2)}(A) \approx \widehat{H}_2^{(2)} = H_2^{(2)} - \eta_h \nabla_{H_2^{(2)}} L_{2,2}^{(tr)}(A, \widehat{W}_2^{(2)}, H_2^{(2)}, D_2^{(\mathrm{tr})}) \quad (18)$$

Finally, we plug the approximations $\widetilde{W}_1^{(2)}(A) \approx \widehat{W}_1^{(2)}$, $\widetilde{H}_1^{(2)}(A) \approx \widehat{H}_1^{(2)}$, $\widetilde{W}_2^{(2)}(A) \approx \widehat{W}_2^{(2)}$, $\widetilde{H}_2^{(2)}(A) \approx \widehat{H}_2^{(2)}$ into the validation loss and get an approximated validation loss. We update $A$ by minimizing the approximated validation loss.

$$A \leftarrow A - \eta_a \nabla_A(L_1^{(val)}(A, \widehat{W}_1^{(2)}, \widehat{H}_1^{(2)}, D_1^{(\mathrm{val})}) + L_2^{(val)}(A, \widehat{W}_2^{(2)}, \widehat{H}_2^{(2)}, D_2^{(\mathrm{val})})) \quad (19)$$

$\nabla_A L_2^{(val)}(A, \widehat{W}_2^{(2)}, \widehat{H}_2^{(2)}, D_2^{(\mathrm{val})})$ can be calculated as:

$$\begin{aligned}
\nabla_A L_2^{(val)}(A, \widehat{W}_2^{(2)}, \widehat{H}_2^{(2)}, D_2^{(\mathrm{val})}) = \\
\frac{\partial L_2^{(val)}(A, \widehat{W}_2^{(2)}, \widehat{H}_2^{(2)}, D_2^{(\mathrm{val})})}{\partial A} + \frac{d\widehat{W}_2^{(2)}}{dA} \frac{\partial L_2^{(val)}(A, \widehat{W}_2^{(2)}, \widehat{H}_2^{(2)}, D_2^{(\mathrm{val})})}{\partial \widehat{W}_2^{(2)}} + \\
\left(\frac{\partial \widehat{H}_2^{(2)}}{\partial A} + \frac{d\widehat{W}_2^{(2)}}{dA} \frac{\partial \widehat{H}_2^{(2)}}{\partial \widehat{W}_2^{(2)}}\right) \frac{\partial L_2^{(val)}(A, \widehat{W}_2^{(2)}, \widehat{H}_2^{(2)}, D_2^{(\mathrm{val})})}{\partial \widehat{H}_2^{(2)}}
\end{aligned} \quad (20)$$

where

$$\frac{d\widehat{W}_2^{(2)}}{dA} = \frac{d\widehat{W}_1^{(2)}}{dA} \frac{\partial \widehat{W}_2^{(2)}}{\partial \widehat{W}_1^{(2)}} + \frac{\partial \widehat{W}_2^{(2)}}{\partial A} \quad (21)$$

where

$$\frac{d\widehat{W}_1^{(2)}}{dA} = \frac{d\widehat{W}_2^{(1)}}{dA} \frac{\partial \widehat{W}_1^{(2)}}{\partial \widehat{W}_2^{(1)}} + \frac{\partial \widehat{W}_1^{(2)}}{\partial A} \quad (22)$$

where

$$\frac{d\widehat{W}_2^{(1)}}{dA} = \frac{d\widehat{W}_1^{(1)}}{dA} \frac{\partial \widehat{W}_2^{(1)}}{\partial \widehat{W}_1^{(1)}} + \frac{\partial \widehat{W}_2^{(1)}}{\partial A} \quad (23)$$

where

$$\frac{d\widehat{W}_1^{(1)}}{dA} = -\eta_w \nabla_{A, W_1^{(1)}}^2 L_{1,1}^{(tr)}(A, W_1^{(1)}, H_1^{(1)}, D_1^{(\mathrm{tr})}) \quad (24)$$

$\nabla_A L_1^{(val)}(A, \widehat{W}_1^{(2)}, \widehat{H}_1^{(2)}, D_1^{(\mathrm{val})})$ can be calculated as:

$$\begin{aligned}
\frac{\partial \nabla_A L_1^{(val)}(A, \widehat{W}_1^{(2)}, \widehat{H}_1^{(2)}, D_1^{(\mathrm{val})})}{\partial A} + \frac{d\widehat{H}_1^{(2)}}{dA} \frac{\partial \nabla_A L_1^{(val)}(A, \widehat{W}_1^{(2)}, \widehat{H}_1^{(2)}, D_1^{(\mathrm{val})})}{\partial \widehat{H}_1^{(2)}} + \\
\frac{d\widehat{W}_1^{(2)}}{dA} \frac{\partial \nabla_A L_1^{(val)}(A, \widehat{W}_1^{(2)}, \widehat{H}_1^{(2)}, D_1^{(\mathrm{val})})}{\partial \widehat{W}_1^{(2)}}
\end{aligned} \quad (25)$$

where

$$\frac{d\widehat{H}_1^{(2)}}{dA} = \frac{\partial \widehat{H}_1^{(2)}}{\partial A} + \frac{d\widehat{W}_1^{(2)}}{dA} \frac{\partial \widehat{H}_1^{(2)}}{\partial \widehat{W}_1^{(2)}} \quad (26)$$

and $\frac{d\widehat{W}_1^{(2)}}{dA}$ is given in Eq.(22).

---

**Algorithm 1** Optimization algorithm for the special case

---

**While** not converged

1. Update the approximation $\widehat{W}_1^{(1)}$ of $\widetilde{W}_1^{(1)}(A)$ using Eq.(11)
2. Update $H_1^{(1)}$ using Eq.(12)
3. Update the approximation $\widehat{W}_2^{(1)}$ of $\widetilde{W}_2^{(1)}(A)$ using Eq.(13)
4. Update the approximation $\widehat{H}_2^{(1)}$ of $\widetilde{H}_2^{(1)}(A)$ using Eq.(14)
5. Update the approximation $\widehat{W}_1^{(2)}$ of $\widetilde{W}_1^{(2)}(A)$ using Eq.(15)
6. Update the approximation $\widehat{H}_1^{(2)}$ of $\widetilde{H}_1^{(2)}(A)$ using Eq.(16)
7. Update the approximation $\widehat{W}_2^{(2)}$ of $\widetilde{W}_2^{(2)}(A)$ using Eq.(17)
8. Update the approximation $\widehat{H}_2^{(2)}$ of $\widetilde{H}_2^{(2)}(A)$ using Eq.(18)
9. Update $A$ using Eq.(19)

---

The gradient descent update of $A$ in Eq.(19) can run one or more steps. After $A$ is updated, the one-step gradient-descent approximations (in equation 2-9), which are functions of $A$, change with $A$ and need to be re-updated. Then, the gradient of $A$, which is a function of one-step gradient-descent approximations, needs to be re-calculated and is used to refresh $A$. In sum, the update of $A$ and the updates of one-step gradient-descent approximations mutually depend on each other. These updates are performed iteratively until convergence. Algorithm 1 shows the algorithm.

In the gradient of $A$ calculated using chain rule, the number of chains is the same as the number of levels in our proposed multi-level optimization (MLO) formulation. This shows that this optimization algorithm preserves the multi-level nested optimization nature of the MLO formulation. In the MLO formulation, $A$ is optimized after finishing $M$ rounds of interleaving learning. In the optimization algorithm, $A$ is updated iteratively, at the end of each of the $M$ rounds. This discrepancy is due to: the optimization of $A$ in the MLO formulation is symbolic, but the update of $A$ in the optimization algorithm is numerical and needs to be conducted iteratively.

## A.2 General Algorithm

Next, we describe the general algorithm that can be applied for any number of tasks and any number of rounds. The special algorithm described in the previous subsection is a special case of the general algorithm introduced in this subsection. Following (Liu et al., 2019a), we approximate $\widetilde{W}_1^{(1)}(A)$ by performing one-step gradient descent update of $W_1^{(1)}$ w.r.t $L(A, W_1^{(1)}, H_1^{(1)}, D_1^{(\mathrm{tr})})$:

$$\widetilde{W}_1^{(1)}(A) \approx \overline{W}_1^{(1)}(A) = W_1^{(1)} - \eta \nabla_{W_1^{(1)}} L(A, W_1^{(1)}, H_1^{(1)}, D_1^{(\mathrm{tr})}). \tag{27}$$

At a middle stage, we approximate $\widetilde{W}_k^{(m)}(A)$ using a one-step gradient descent update of $W_k^{(m)}$ w.r.t. $\sum_{\mathcal{A}_i, \mathcal{A}_j} \max\left(0, -\big(d(\mathcal{A}_i, \mathcal{A}_j; \overline{W}_{k-1}^{(m)}(A)) - \tau\big)\big(d(\mathcal{A}_i, \mathcal{A}_j; W_k^{(m)}) - \tau\big)\right)$, where $\overline{W}_{k-1}^{(m)}(A)$ is the approximation of $\widetilde{W}_{k-1}^{(m)}(A)$.

$$\begin{aligned}
\widetilde{W}_k^{(m)}(A) \approx \overline{W}_k^{(m)}(A) &= W_k^{(m)}(A) \\
&- \eta \nabla_{W_k^{(m)}(A)} \left(\sum_{\mathcal{A}_i, \mathcal{A}_j} \max\left(0, -\big(d(\mathcal{A}_i, \mathcal{A}_j; \overline{W}_{k-1}^{(m)}(A)) - \tau\big)\big(d(\mathcal{A}_i, \mathcal{A}_j; W_k^{(m)}) - \tau\big)\right)\right).
\end{aligned} \tag{28}$$

Note that $\{\overline{W}_k^{(m)}(A)\}_{k,m=1}^{K,M}$ are calculated recursively, where $\overline{W}_k^{(m)}(A)$ is a function of $\overline{W}_{k-1}^{(m)}(A)$, $\overline{W}_{k-1}^{(m)}(A)$ is a function of $\overline{W}_{k-2}^{(m)}(A)$, and so on. When $m > 1$ and $k = 1$, $\overline{W}_{k-1}^{(m)}(A) = \overline{W}_K^{(m-1)}(A)$. For $\widetilde{H}_k^{(m)}(A)$, the approximation is:

$$\overline{H}_k^{(m)}(A) = H_k^{(m)}(A) - \eta \nabla_{H_k^{(m)}(A)} L(A, \overline{W}_k^{(m)}, H_k^{(m)}, D_k^{(\mathrm{tr})}). \tag{29}$$

In the validation stage, we plug the approximations of $\{\widetilde{W}_k^{(M)}(A)\}_{k=1}^K$ and $\{\widetilde{H}_k^{(M)}(A)\}_{k=1}^K$ into the validation loss function, calculate the gradient of the approximated objective w.r.t the encoder architecture $A$, then update $A$ via:

$$A \leftarrow A - \eta \sum_{k=1}^K \nabla_A L(A, \overline{W}_k^{(M)}(A), \overline{H}_k^{(M)}(A), D_k^{(\text{val})}). \tag{30}$$

We reiterate the analysis given before. After $A$ is updated in equation 21, the one-step gradient-descent approximations (in equation 18-20), which are functions of $A$, need to be re-updated. The gradient update of $A$, which is a function of one-step gradient-descent approximations, needs to be re-calculated. In sum, the update of $A$ and the updates of one-step gradient-descent approximations mutually depend on each other. The update steps from Eq.(27) to Eq.(30) iterate until convergence. The entire algorithm is summarized in Algorithm 2.

---

**Algorithm 2** Optimization algorithm for interleaving multi-task learning

---

**While** not converged

    Update $\widetilde{W}_1^{(1)}(A)$ using Eq.(27)

   **For** (m=1 and k=2:K) and (m=2:M and k=1:K)

       Update $\widetilde{W}_k^{(m)}(A)$ using Eq.(28)

       Update $\widetilde{H}_k^{(m)}(A)$ using Eq.(29)

   **End**

    Update $A$ using Eq.(30)

---

## A.3 Pseudo Code

The pseudo code of our method is shown below.

**While** not converged

#Update $W_1^{(1)}$ and $H_1^{(1)}$
Sample a minibatch $m_1$ from $D_1^{(\text{tr})}$
Calculate gradient $\triangle W_1^{(1)} = \nabla_{W_1^{(1)}} L_{1,1}^{(tr)}(A, W_1^{(1)}, H_1^{(1)}, m_1)$
Update $W_1^{(1)} \leftarrow W_1^{(1)} - \eta_w \triangle W_1^{(1)}$
Calculate gradient $\triangle H_1^{(1)} = \nabla_{H_1^{(1)}} L_{1,1}^{(tr)}(A, W_1^{(1)}, H_1^{(1)}, m_1)$
Update $H_1^{(1)} \leftarrow H_1^{(1)} - \eta_h \triangle H_1^{(1)}$

#Update $\{W_k^{(1)}, H_k^{(1)}\}_{k=2}^K$
For $k = 2 : K$
    Sample ten examples from $D_{k-1}^{(\text{tr})}$ and ten examples from $D_k^{(\text{tr})}$
    For each example, construct an augmented set $\mathcal{A}_i$
    Calculate gradient $\triangle W_k^{(1)} = \nabla_{W_k^{(1)}} \left( \sum_{\mathcal{A}_i, \mathcal{A}_j} \max\left( 0, -\left(d(\mathcal{A}_i, \mathcal{A}_j; W_{k-1}^{(1)}) - \tau\right)\left(d(\mathcal{A}_i, \mathcal{A}_j; W_k^{(1)}) - \tau\right)\right)\right)$
    Update $W_k^{(1)} \leftarrow W_k^{(1)} - \eta_w \triangle W_k^{(1)}$
    Sample a minibatch $m_k$ from $D_k^{(\text{tr})}$
    Calculate gradient $\triangle H_k^{(1)} = \nabla_{H_k^{(1)}} L(A, W_k^{(1)}, H_k^{(1)}, m_k)$
    Update $H_k^{(1)} \leftarrow H_k^{(1)} - \eta_h \triangle H_k^{(1)}$
End

#Update $\{W_k^{(m)}, H_k^{(m)}\}_{m=2, k=1}^{M, K}$
For $m = 2 : M, \ k = 1 : K$
    Sample ten examples from $D_{k-1}^{(\text{tr})}$ and ten examples from $D_k^{(\text{tr})}$
    For each example, construct an augmented set $\mathcal{A}_i$
    Calculate gradient $\triangle W_k^{(m)} = \nabla_{W_k^{(m)}} \left( \sum_{\mathcal{A}_i, \mathcal{A}_j} \max\left( 0, -\left(d(\mathcal{A}_i, \mathcal{A}_j; W_{k-1}^{(m)}) - \tau\right)\left(d(\mathcal{A}_i, \mathcal{A}_j; W_k^{(m)}) - \tau\right)\right)\right)$
    Update $W_k^{(m)} \leftarrow W_k^{(m)} - \eta_w \triangle W_k^{(m)}$
    Sample a minibatch $m_k$ from $D_k^{(\text{tr})}$
    Calculate gradient $\triangle H_k^{(m)} = \nabla_{H_k^{(m)}} L(A, W_k^{(m)}, H_k^{(m)}, D_k^{(\text{tr})})$
    Update $H_k^{(m)} \leftarrow H_k^{(m)} - \eta_h \triangle H_k^{(m)}$
End

#Update $A$
For $k = 1 : K$
    Sample a minibatch $m_k$ from $D_k^{(\text{tr})}$
End
Calculate gradient $\triangle A = \sum_{k=1}^K \nabla_A L(A, W_k^{(M)}, H_k^{(M)}, m_k)$
Update $A \leftarrow A - \eta_a \triangle A$

End

# B   Experiments of IMTNAS on three tasks

In this section, we perform an additional experiment, which performs IMTNAS on three tasks in a unified framework: image classification on CIFAR-10, image classification on CIFAR-100, and image classification on ImageNet, where classification on ImageNet is the third learning task. For CIFAR-10 and CIFAR-100, the experimental settings are the same as those described in Section 4.1.2 in the main paper. For ImageNet, following (Xu et al., 2020), we randomly sample 10% of the 1.3 training images as training sets and randomly sample another 2.5% of the 1.3 training images as validation sets. The classification head for ImageNet is a 1000-way linear classifier. ImageNet shares the same encoder architecture as CIFAR-10 and CIFAR-100. We set the number of interleaving rounds to 2. The tradeoff parameter $\lambda$ is set to 100. IMTNAS is applied to PC-DARTS in consideration of its computational efficiency. The order of tasks in the interleaving process is: CIFAR-100, CIFAR-10, ImageNet, CIFAR-100, CIFAR-10, ImageNet. The rest of hyperparameters are the same as those described in Section 4.1.2 in the main paper.

Table 11, 12, and 13 shows the results on the three datasets. As can be seen, our IMTNAS method outperforms PC-DARTS and MTL-SWS. These results further demonstrate the effectiveness of interleaving learning. The performance of DARTS+ is unstable across different datasets. For example, as shown in Table 12, the test error of DARTS+ on CIFAR-10 is 2.83±0.05, which is much worse than the 2.49±0.04 error rate of our method. In contrast, our approach consistently achieves the best performance across different datasets.

We also experimented with different orders on the three tasks. Table 14 shows the results. Similar to Table 7, task order does not affect performance significantly.

Table 11: Classification errors on the test set of CIFAR-100, number of model parameters, and search cost (GPU days). IMTNAS-DARTS2nd denotes that our proposed IMTNAS framework is applied to the search space of DARTS-2nd. DARTS-1st and DARTS-2nd means that first order and second order approximation is used in DARTS' optimization procedure. Results marked with * are taken from DARTS$^-$ (Chu et al., 2020a). Methods marked with † were re-run for 10 times with different random initialization. $\Delta$ denotes this algorithm ran for 600 epochs instead of 2000 epochs in the architecture evaluation stage, to ensure the comparison with other methods (which all ran for 600 epochs) is fair.

| Method | Error(%) | Param(M) | Cost |
|---|---|---|---|
| *ResNet (He et al., 2016) | 22.10 | 1.7 | - |
| *DenseNet (Huang et al., 2017) | 17.18 | 25.6 | - |
| *PNAS (Liu et al., 2018a) | 19.53 | 3.2 | 150 |
| *ENAS (Pham et al., 2018) | 19.43 | 4.6 | 0.5 |
| *AmoebaNet (Real et al., 2019) | 18.93 | 3.1 | 3150 |
| †DARTS-1st (Liu et al., 2019a) | 20.52±0.31 | 1.8 | 0.4 |
| *GDAS (Dong & Yang, 2019) | 18.38 | 3.4 | 0.2 |
| *R-DARTS (Zela et al., 2020) | 18.01±0.26 | - | 1.6 |
| *DARTS$^-$ (Chu et al., 2020a) | 17.51±0.25 | 3.3 | 0.4 |
| †DARTS$^-$ (Chu et al., 2020a) | 18.97±0.16 | 3.1 | 0.4 |
| $\Delta$DARTS$^+$ (Liang et al., 2019) | 17.11±0.43 | 3.8 | 0.2 |
| *DropNAS (Hong et al., 2020b) | 16.39 | 4.4 | 0.7 |
| †PC-DARTS (Xu et al., 2020) | 17.96±0.15 | 3.9 | 0.1 |
| MTLSWS-PCDARTS | 17.93±0.21 | 3.9 | 0.3 |
| IMTNAS-PCDARTS (ours) | **17.15**±0.04 | 4.1 | 0.6 |

# C   Additional results of baselines

Table 15 shows results of additional baselines in the ImageNet-CoCo interleaving experiment. Table 16 shows results of additional baselines in the Cifar100-Cifar10 interleaving experiment.

# D   Definition of MMD

Let $k(\cdot, \cdot)$ be a kernel function. Given two distributions $p$ and $q$, their MMD is defined as:

$$\mathbb{E}_{x \sim p, x' \sim p}[k(x, x')] + \mathbb{E}_{y \sim q, y' \sim q}[k(y, y')] - 2\mathbb{E}_{x \sim p, y \sim q}[k(x, y)]. \tag{31}$$

Table 12: Classification errors on the test set of CIFAR-10, number of model parameters, and search cost. Results marked with * are taken from DARTS$^-$ (Chu et al., 2020a), NoisyDARTS (Chu et al., 2020b), and DrNAS (Chen et al., 2020b). The rest notations are the same as those in Table 11.

| Method | Error(%) | Param(M) | Cost |
|---|---|---|---|
| *DenseNet (Huang et al., 2017) | 3.46 | 25.6 | - |
| *HierEvol (Liu et al., 2018b) | 3.75±0.12 | 15.7 | 300 |
| *NAONet-WS (Luo et al., 2018) | 3.53 | 3.1 | 0.4 |
| *PNAS (Liu et al., 2018a) | 3.41±0.09 | 3.2 | 225 |
| *ENAS (Pham et al., 2018) | 2.89 | 4.6 | 0.5 |
| *NASNet-A (Zoph et al., 2018) | 2.65 | 3.3 | 1800 |
| *AmoebaNet-B (Real et al., 2019) | 2.55±0.05 | 2.8 | 3150 |
| *DARTS-1st (Liu et al., 2019a) | 3.00±0.14 | 3.3 | 0.4 |
| *R-DARTS (Zela et al., 2020) | 2.95±0.21 | - | 1.6 |
| *GDAS (Dong & Yang, 2019) | 2.93 | 3.4 | 0.2 |
| *SNAS (Xie et al., 2019) | 2.85 | 2.8 | 1.5 |
| $^\triangle$DARTS$^+$ (Liang et al., 2019) | 2.83±0.05 | 3.7 | 0.4 |
| *BayesNAS (Zhou et al., 2019) | 2.81±0.04 | 3.4 | 0.2 |
| *MergeNAS (Wang et al., 2020a) | 2.73±0.02 | 2.9 | 0.2 |
| *NoisyDARTS (Chu et al., 2020b) | 2.70±0.23 | 3.3 | 0.4 |
| *ASAP (Noy et al., 2020) | 2.68±0.11 | 2.5 | 0.2 |
| *SDARTS (Chen & Hsieh, 2020) | 2.61±0.02 | 3.3 | 1.3 |
| *DARTS$^-$ (Chu et al., 2020a) | 2.59±0.08 | 3.5 | 0.4 |
| $^\dagger$DARTS$^-$ (Chu et al., 2020a) | 2.97±0.04 | 3.3 | 0.4 |
| *DropNAS (Hong et al., 2020b) | 2.58±0.14 | 4.1 | 0.6 |
| *FairDARTS (Chu et al., 2019b) | 2.54 | 3.3 | 0.4 |
| *DrNAS (Chen et al., 2020b) | 2.54±0.03 | 4.0 | 0.4 |
| *PC-DARTS (Xu et al., 2020) | 2.57±0.07 | 3.6 | 0.1 |
| MTLSWS-PCDARTS | 2.61±0.07 | 3.9 | 0.3 |
| IMTNAS-PCDARTS (ours) | **2.49**±0.04 | 4.0 | 0.6 |

Given a set of sample $\{x_i\}_{i=1}^m$ drawn from $p$ and a set of sample $\{y_i\}_{i=1}^n$ drawn from $q$, the empirical MMD is calculated as:

$$\frac{1}{m(m-1)} \sum_{i=1}^m \sum_{j \neq i}^m k(x_i, x_j) + \frac{1}{n(n-1)} \sum_{i=1}^n \sum_{j \neq i}^n k(y_i, y_j) - \frac{2}{mn} \sum_{i=1}^m \sum_{j=1}^n k(x_i, y_j). \tag{32}$$

# E  Experimental details of neural architecture search

## E.1  DARTS2nd based experiments

For methods based on DARTS2nd, including IMTNAS-Darts2nd, SI-Darts2nd, MI-Darts2nd, MTL-SWS-Darts2nd, MTL-HWS-Darts2nd, MTL-SAO-Darts2nd, the experimental settings are similar. In search spaces of DARTS, the candidate operations include: $3 \times 3$ and $5 \times 5$ separable convolutions, $3 \times 3$ and $5 \times 5$ dilated separable convolutions, $3 \times 3$ max pooling, $3 \times 3$ average pooling, identity, and zero. The network is a stack of multiple cells, each consisting of 7 nodes. The stride of all operations is set to 1. The convolved feature maps are padded to preserve their spatial resolution. The order for convolutional operations is ReLU-Conv-BN. Each separable convolution is applied twice. The convolutional cell has 7 nodes. The output node is the depthwise concatenation of all intermediate nodes, excluding the input nodes. We create a network by stacking 8 cells. The first and second nodes of cell $k$ are equal to the outputs of cell $k-2$ and cell $k-1$, respectively. 1×1 convolutions are inserted when necessary. Reduction cells are located at the 1/3 and 2/3 of the total depth of the network. In reduction cells, operations adjacent to the input nodes have a stride of 2.

For CIFAR-10 and CIFAR-100, during architecture search, the encoder architecture is a stack of 8 cells, each consisting of 7 nodes, with the initial channel number set to 16. Network weights were optimized using the SGD optimizer with a batch size of 64, an initial learning rate of 0.025, a learning rate scheduler of cosine decay, a weight decay of 3e-4, a momentum of 0.9, and an epoch number of 50. For regular Darts, the number of epochs is set to 150. The architecture variables were optimized using the Adam (Kingma & Ba, 2014) optimizer with a learning rate of 3e-4, a momentum of $(0.5, 0.999)$, and a weight decay of 1e-3. The learning rate was scheduled with cosine scheduling. The architecture variables were initialized with zero initialization.

Table 13: Top-1 and top-5 classification errors on the test set of ImageNet, number of model parameters, and search cost (GPU days). Results marked with * were taken from DARTS⁻ (Chu et al., 2020a) and DrNAS (Chen et al., 2020b). The rest notations are the same as those in Table 11. From top to bottom, on the first, second, and third block are: 1) networks manually designed by humans; 2) non-differentiable architecture search methods; and 3) differentiable search methods.

| Method | Top-1 Error (%) | Top-5 Error (%) | Param (M) | Cost (GPU days) |
|---|---|---|---|---|
| *Inception-v1 (Szegedy et al., 2015) | 30.2 | 10.1 | 6.6 | - |
| *MobileNet (Howard et al., 2017) | 29.4 | 10.5 | 4.2 | - |
| *ShuffleNet 2× (v1) (Zhang et al., 2018) | 26.4 | 10.2 | 5.4 | - |
| *ShuffleNet 2× (v2) (Ma et al., 2018) | 25.1 | 7.6 | 7.4 | - |
| *NASNet-A (Zoph et al., 2018) | 26.0 | 8.4 | 5.3 | 1800 |
| *PNAS (Liu et al., 2018a) | 25.8 | 8.1 | 5.1 | 225 |
| *MnasNet-92 (Tan et al., 2019a) | 25.2 | 8.0 | 4.4 | 1667 |
| *AmoebaNet-C (Real et al., 2019) | 24.3 | 7.6 | 6.4 | 3150 |
| *SNAS (Xie et al., 2019) | 27.3 | 9.2 | 4.3 | 1.5 |
| *BayesNAS (Zhou et al., 2019) | 26.5 | 8.9 | 3.9 | 0.2 |
| *PARSEC (Casale et al., 2019) | 26.0 | 8.4 | 5.6 | 1.0 |
| *GDAS (Dong & Yang, 2019) | 26.0 | 8.5 | 5.3 | 0.2 |
| *DSNAS (Hu et al., 2020) | 25.7 | 8.1 | - | - |
| *SDARTS-ADV (Chen & Hsieh, 2020) | 25.2 | 7.8 | 5.4 | 1.3 |
| *PC-DARTS (Xu et al., 2020) | 25.1 | 7.8 | 5.3 | 0.1 |
| *ProxylessNAS (Cai et al., 2019) | 24.9 | 7.5 | 7.1 | 8.3 |
| *FairDARTS (CIFAR-10) (Chu et al., 2019b) | 24.9 | 7.5 | 4.8 | 0.4 |
| *FairDARTS (ImageNet) (Chu et al., 2019b) | 24.4 | 7.4 | 4.3 | 3.0 |
| *DrNAS (Chen et al., 2020b) | 24.2 | 7.3 | 5.2 | 3.9 |
| *DARTS⁺ (ImageNet) (Liang et al., 2019) | 23.9 | 7.4 | 5.1 | 6.8 |
| *DARTS⁻ (Chu et al., 2020a) | 23.8 | 7.0 | 4.9 | 4.5 |
| *DARTS⁺ (CIFAR-100) (Liang et al., 2019) | 23.7 | 7.2 | 5.1 | 0.2 |
| *PC-DARTS (Xu et al., 2020) | 24.2 | 7.3 | 5.3 | 3.8 |
| MTLSWS-PCDARTS | 24.4 | 7.6 | 5.5 | 6.8 |
| IMTNAS-PCDARTS (ours) | **23.7** | **6.8** | 5.5 | 7.0 |

Table 14: Ablation study on task order.

| Order | ImageNet top-1 error | CIFAR-100 error | CIFAR-10 error |
|---|---|---|---|
| CIFAR-100, CIFAR-10, ImageNet, CIFAR-100, CIFAR-10, ImageNet | 23.7 | 17.15±0.04 | 2.49±0.04 |
| CIFAR-100, ImageNet, CIFAR-10, CIFAR-100, ImageNet, CIFAR-10 | 23.6 | 17.18±0.06 | 2.47±0.04 |
| ImageNet, CIFAR-10, CIFAR-100, ImageNet, CIFAR-10, CIFAR-100 | 23.9 | 17.13±0.03 | 2.46±0.05 |

Table 15: Object detection on COCO test set and classification accuracy on ImageNet test set. APS, APM, APL: AP at small, medium, large scale. AP50 and AP75 are AP with IoU thresholds of 0.5 and 0.75. Acc is accuracy.

| | COCO | | | | | | | ImageNet | | ×+ |
|---|---|---|---|---|---|---|---|---|---|---|
| | Acc | AP | AP50 | AP75 | APS | APM | APL | Top-1 | Top-5 | (M) |
| MobileNetV2 (Sandler et al., 2018) | 72.0 | 28.3 | 46.7 | 29.3 | 14.8 | 30.7 | 38.1 | 72.0 | 91.0 | 300 |
| FairNAS-A (Chu et al., 2019a) | 77.5 | 32.4 | 52.4 | 33.9 | 17.2 | 36.3 | 43.2 | 77.5 | 93.7 | 392 |
| FairNAS-B (Chu et al., 2019a) | 77.2 | 31.7 | 51.5 | 33.0 | 17.0 | 35.2 | 42.5 | 77.2 | 93.5 | 349 |
| IMTNAS (ours) | **77.8** | **32.7** | **53.3** | **34.1** | **17.2** | **37.1** | **43.8** | **77.8** | **94.0** | 394 |

Given the optimally searched architecture cell, we evaluate it individually on CIFAR-10, CIFAR-100, and ImageNet. For CIFAR-10 and CIFAR-100, we stack 20 copies of the searched cell into a larger network as the image encoder. The initial channel number was set to 36. We trained the network for 600 epochs on the combination of the training and validation datasets where the mini-batch size was set to 96. For ImageNet, similar to (Liu et al., 2019a), we evaluate the architecture cells searched on CIFAR10/100. A larger network is formed by stacking 14 copies of the searched cell. The initial channel number was set to 48. We trained the network for 250 epochs on the 1.2M training images

Table 16: Classification errors (%) on test sets of CIFAR-100 and CIFAR-10, number of model parameters (millions), and search cost (GPU days on a Nvidia 1080Ti, sum of costs of all tasks). Results on ImageNet are under mobile settings. IMTNAS-Darts2nd: our IMTNAS is applied to DARTS-2nd. Results marked with * are taken from DARTS$^-$ (Chu et al., 2020a). Methods marked with † were re-run 10 times with random initialization.

| | CIFAR100 Error(%) | CIFAR10 Error(%) | ImageNet Top-1 | Top-5 | Param (M) | Cost (GPU days) |
|---|---|---|---|---|---|---|
| *ResNet (He et al., 2016) | 22.10 | 6.43 | - | - | 1.7 | - |
| *DenseNet (Huang et al., 2017) | 17.18 | 3.46 | - | - | 25.6 | - |
| *PNAS (Liu et al., 2018a) | 19.53 | 3.41±0.09 | 25.8 | 8.1 | 3.2 | 150 |
| *ENAS (Pham et al., 2018) | 19.43 | 2.89 | - | - | 4.6 | 0.5 |
| *AmoebaNet (Real et al., 2019) | 18.93 | 2.55±0.05 | 24.3 | 7.6 | 3.1 | 3150 |
| †DARTS-1st (Liu et al., 2019a) | 20.52±0.31 | 3.00±0.14 | - | - | 1.8 | 0.4 |
| *GDAS (Dong & Yang, 2019) | 18.38 | 2.93 | 26.0 | 8.5 | 3.4 | 0.2 |
| *DARTS$^-$ (Chu et al., 2020a) | 17.51±0.25 | 2.59±0.08 | 23.8 | 7.0 | 3.3 | 0.4 |
| *DropNAS (Hong et al., 2020b) | 16.95±0.41 | 2.58±0.14 | 23.4 | 6.7 | 4.4 | 0.7 |

where the batch size was set to 1024. Other hyperparameters are the same as those in architecture search. Cutout, path dropout of probability 0.2 and auxiliary towers with weight 0.4 were applied.

## E.2 PC-DARTS based experiments

For methods based on PC-DARTS, including IMTNAS-pcdarts (ours), SI-pcdarts, MI-pcdarts, MTL-SWS-pcdarts, MTL-HWS-pcdarts, MTL-SAO-pcdarts, the experimental settings are similar. The search space of PC-DARTS follows that of DARTS. For architecture search on CIFAR-100 and CIFAR-10, the hyperparameter $K$ was set to 4. The network is a stack of 8 cells. Each cell contains 6 nodes. Initial channel number is set to 16. The architecture variables are trained using the Adam optimizer. The learning rate is set to $6e-4$, without decay. The weight decay is set to $1e-3$. The momentum is set to $(0.5, 0.999)$. The network weight parameters are trained using SGD. The initial learning rate is set to 0.1. Cosine scheduling is used to decay the learning rate, down to 0 without restart. The momentum is set to 0.9. The weight decay is set to $3e-4$. The batch size is set to 256. Warm-up is utilized: in the first 15 epochs, architecture variables are frozen and only network weights are optimized.

The settings for architecture evaluation on CIFAR-100 and CIFAR-10 follow those of DARTS. 18 normal cells and 2 reduction cells are stacked into a large network. The initial channel number is set to 36. The stacked network is trained from scratch using SGD for 600 epochs, with batch size 128, initial learning rate 0.025, momentum 0.9, weight decay $3e-4$, norm gradient clipping 5, drop-path rate 0.3, and cutout. The learning rate is decayed to 0 using cosine scheduling without restart.

We combine our method and PC-DARTS to directly search for architectures on ImageNet. The stacked network starts with three convolution layers which reduce the input image resolution from 224×224 to 28×28, using stride 2. After the three convolution layers, 6 normal cells and 2 reduction cells are stacked. Each cell consists of $N = 6$ nodes. The sub-sampling rate was set to 0.5. Architecture variables are trained using Adam. The learning rate is fixed to $6e-3$. The weight decay is set to $1e-3$. The momentum is set to $(0.5, 0.999)$. In the first 35 epochs, architecture variables are frozen. Network weight parameters are trained using SGD. The initial learning rate is set to 0.5. It is decayed to 0 using cosine scheduling without restart. Momentum is set to 0.9. Weight decay is set to $3e-5$. The batch-size is set to 1024. Epoch number is set to 250. Eight Tesla V100 GPUs were used.

For architecture evaluation on ImageNet, the stacked network starts with three convolution layers which reduce the input image resolution from 224×224 to 28×28, using stride 2. After the three convolution layers, 12 normal cells and 2 reduction cells are stacked. Initial channel number is set to 48. The network is trained from scratch using SGD for 250 epochs, with batch size 1024, initial learning rate 0.5, weight decay $3e-5$, and momentum 0.9. For the first 5 epochs, learning rate warm-up is used. The learning rate is linearly decayed to 0. Label smoothing and auxiliary loss tower is used.

## E.3 P-DARTS based experiments

The search process has three stages. At the first stage, the search space and stacked network in P-DARTS are mostly the same as DARTS. The only difference is the number of cells in the stacked network in P-DARTS is set to 5. At the

second stage, the number of cells in the stacked network is 11. At the third stage, the cell number is 17. At stage 1, 2, 3, the initial Dropout probability on skip-connect is 0, 0.4, and 0.7 for CIFAR-10, is 0.1, 0.2, and 0.3 for CIFAR-100; the size of operation space is 8, 5, 3, respectively. The final searched cell is limited to have 2 skip-connect operations at maximum. At each stage, the network is trained using the Adam optimizer. The batch size is set to 96. The learning rate is set to 6e-4. Weight decay is set to 1e-3. Momentum is set to $(0.5, 0.999)$. In the first 10 epochs, architecture variables are frozen and only network weights are optimized.

For architecture evaluation on CIFAR-100 and CIFAR-10, the stacked network consists of 20 cells. The initial channel number is set to 36. The network is trained from scratch using SGD. The epoch number is set to 600. The batch size is set to 128. The initial learning rate is set to 0.025. The learning rate is decayed to 0 using cosine scheduling. Weight decay is set to 3e-4 for CIFAR-10 and 5e-4 for CIFAR-100. Momentum is set to 0.9. Drop-path probability is set to 0.3. Cutout regularization length is set to 16. Auxiliary towers of weight 0.4 are used.

For architecture evaluation on ImageNet, the settings are similar to those of DARTS. The network consists of 14 cells. The initial channel number is set to 48. The network is trained from scratch using SGD for 250 epochs. Batch size is set to 1024. Initial learning rate is set to 0.5. The learning rate is linearly decayed after each epoch. In the first 5 epochs, learning rate warmup is used. The momentum is set to 0.9. The weight decay is set to $3e-5$. Label smoothing and auxiliary loss tower are used during training. The network was trained on 8 Nvidia Tesla V100 GPUs.

### E.4    PR-DARTS based experiments

The operations include: $3\times3$ and $5\times5$ separable convolutions, $3\times3$ and $5\times5$ dilated separable convolutions, $3\times3$ average pooling and $3\times3$ max pooling, zero, and skip connection. The stacked network consists of $k$ cells. The $k/3$- and $2k/3$-th cells are reduction cells. In reduction cells, all operations have a stride of two. The rest cells are normal cells. Operations in normal cells have a stride of one. Cells of the same type (either reduction or normal) have the same architecture. The inputs of each cell are the outputs of two previous cells. Each cell contains four intermediate nodes and one output node. The output node is a concatenation of all intermediate nodes.

For architecture search on CIFAR-100 and CIFAR-10, the stacked network consists of 8 cells. The initial channel number is set to 16. In PR-DARTS, $\lambda_1$, $\lambda_2$, and $\lambda_3$ are set to 0.01, 0.005, and 0.005 respectively. The network was trained for 200 epochs. The mini-batch size is set to 128. Architecture variables are trained using Adam. The learning rate is set to $3e-4$. The weight decay is set to $1e-3$. Network weights are trained using SGD. The initial learning rate is set to 0.025. The momentum is set to 0.9. The weight decay is set to $3e-4$. The learning rate is decayed to 0 using cosine scheduling. For acceleration, per iteration, only two operations on each edge are randomly selected to update. The temperature $\tau$ is set to 10 and is linearly reduced to 0.1; $a = -0.1$ and $b = 1.1$. Pruning on each node is conducted by comparing the gate activation probabilities of all non-zero operations collected from all previous nodes and retaining top two operations.

For architecture evaluation on CIFAR10 and CIFAR100, the stacked network consists of 18 normal cells and 2 reduction cells. The initial channel number is set to 36. The network is trained from scratch using SGD. The mini-batch size is set to 128. The epoch number is set to 600. The initial learning rate is set to 0.025. The momentum is set to 0.9. The weight decay is set to $3e-4$. The gradient norm clipping is set to 5. The drop-path probability is set to 0.2. The cutout length is set to 16. The learning rate is decayed to 0 using cosine scheduling.

For architecture evaluation on ImageNet, the input images are resized to $224\times224$. The stacked network consists of 3 convolutional layers, 12 normal cells, and 2 reduction cells. The channel number is set to 48. The network is trained using SGD for 250 epochs. The batch size is set to 128. The learning rate is set to 0.025. The momentum is set to 0.9. The weight decay is set to $3e-4$. The gradient norm clipping is set to 5. The learning rate is decayed to 0 via cosine scheduling.

### E.5    Hyperparameter tuning strategy

To tune the interleaving round number $K$, we randomly sample 2.5K data from the 25K training set and sample 2.5K data from the 25K validation set. Then we use the 5K sampled data as a hyperparameter tuning set. $K$ is tuned in $\{2, 3, 4\}$. For each configuration of $K$, we use the remaining 22.5K training data and 22.5K validation data to perform architecture search and use their combination to perform architecture evaluation (retraining a larger stacked network from scratch). Then we measure the performance of the stacked network on the 5K sampled data. The value of $K$

| Our method | Baseline | p-value |
|---|---|---|
| IMTNAS-Darts2nd | SI-Darts2nd | 8.16e-12 |
| IMTNAS-Darts2nd | MI-Darts2nd | 2.83e-11 |
| IMTNAS-Darts2nd | MTL-SWS-Darts2nd | 4.29e-9 |
| IMTNAS-Darts2nd | MTL-HWS-Darts2nd | 4.48e-11 |
| IMTNAS-Darts2nd | MTL-SAO-Darts2nd | 1.25e-11 |
| IMTNAS-Darts2nd | Darts2nd | 5.60e-14 |
| IMTNAS-Pdarts | SI-Pdarts | 6.21e-7 |
| IMTNAS-Pdarts | MI-Pdarts | 9.75e-7 |
| IMTNAS-Pdarts | MTL-SWS-Pdarts | 3.62e-6 |
| IMTNAS-Pdarts | MTL-HWS-Pdarts | 5.65e-7 |
| IMTNAS-Pdarts | MTL-SAO-Pdarts | 1.01e-7 |
| IMTNAS-Pdarts | Pdarts | 4.49e-7 |
| IMTNAS-Pcdarts | SI-Pcdarts | 3.25e-3 |
| IMTNAS-Pcdarts | MI-Pcdarts | 6.50e-4 |
| IMTNAS-Pcdarts | MTL-SWS-Pcdarts | 9.48e-4 |
| IMTNAS-Pcdarts | MTL-HWS-Pcdarts | 1.33e-5 |
| IMTNAS-Pcdarts | MTL-SAO-Pcdarts | 7.06e-5 |
| IMTNAS-Pcdarts | Pcdarts | 5.11e-3 |
| IMTNAS-Prdarts | SI-Prdarts | 1.87e-9 |
| IMTNAS-Prdarts | MI-Prdarts | 4.36e-10 |
| IMTNAS-Prdarts | MTL-SWS-Prdarts | 9.25e-7 |
| IMTNAS-Prdarts | MTL-HWS-Prdarts | 2.72e-8 |
| IMTNAS-Prdarts | MTL-SAO-Prdarts | 8.77e-9 |
| IMTNAS-Prdarts | Prdarts | 2.29e-5 |

Table 17: Significance test results on CIFAR-100

yielding the best performance on the 5K sampled data is selected. For other hyperparameters, they mostly follow those in DARTS (Liu et al., 2019a), P-DARTS (Chen et al., 2019), PC-DARTS (Xu et al., 2020), and PR-DARTS (Zhou et al., 2020).

### E.6    Implementation details

We use PyTorch to implement all models. The version of Torch is 1.4.0 (or above). We build our method upon official python packages for different differentiable search approaches, such as "DARTS[1]", "P-DARTS[2]" and "PC-DARTS[3]".

## F    Significance test results

To check whether the performance of our proposed methods are significantly better than baselines, we perform a statistical significance test using a double-sided T-test. We use the function in the python package "scipy.stats.ttest_1samp" and report the average results over 10 different runs. Table 17 and 18 show the results.

From these two tables, we can see that the p-values are small between baselines methods and our methods, which demonstrate that the errors of our methods are significantly lower than those of baselines.

---

[1]https://github.com/quark0/darts
[2]https://github.com/chenxin061/pdarts
[3]https://github.com/yuhuixu1993/PC-DARTS/

| Our method | Baseline | p-value |
|---|---|---|
| IMTNAS-Darts2nd | SI-Darts2nd | 8.66e-8 |
| IMTNAS-Darts2nd | MI-Darts2nd | 4.01e-7 |
| IMTNAS-Darts2nd | MTL-SWS-Darts2nd | 9.73e-5 |
| IMTNAS-Darts2nd | MTL-HWS-Darts2nd | 9.95e-7 |
| IMTNAS-Darts2nd | MTL-SAO-Darts2nd | 2.29e-7 |
| IMTNAS-Darts2nd | Darts2nd | 4.06e-5 |
| IMTNAS-Pdarts | SI-Pdarts | 4.81e-7 |
| IMTNAS-Pdarts | MI-Pdarts | 6.29e-7 |
| IMTNAS-Pdarts | MTL-SWS-Pdarts | 8.78e-6 |
| IMTNAS-Pdarts | MTL-HWS-Pdarts | 3.35e-7 |
| IMTNAS-Pdarts | MTL-SAO-Pdarts | 1.75e-7 |
| IMTNAS-Pdarts | Pdarts | 3.84e-6 |
| IMTNAS-Pcdarts | SI-Pcdarts | 1.92e-8 |
| IMTNAS-Pcdarts | MI-Pcdarts | 5.60e-9 |
| IMTNAS-Pcdarts | MTL-SWS-Pcdarts | 7.66e-6 |
| IMTNAS-Pcdarts | MTL-HWS-Pcdarts | 8.04e-9 |
| IMTNAS-Pcdarts | MTL-SAO-Pcdarts | 2.95e-8 |
| IMTNAS-Pcdarts | Pcdarts | 1.34e-6 |
| IMTNAS-Prdarts | SI-Prdarts | 7.29e-10 |
| IMTNAS-Prdarts | MI-Prdarts | 9.05e-11 |
| IMTNAS-Prdarts | MTL-SWS-Prdarts | 2.36e-8 |
| IMTNAS-Prdarts | MTL-HWS-Prdarts | 6.51e-9 |
| IMTNAS-Prdarts | MTL-SAO-Prdarts | 2.77e-9 |
| IMTNAS-Prdarts | Prdarts | 4.26e-3 |

Table 18: Significance test results on CIFAR-10

# G  Runtime, validation performance, number of weight parameters, implementation details

## G.1  For architecture search

The runtime (seconds per epoch) on architecture search is shown in Table 19. The validation performance is shown in Table 20 and 21. We use PyTorch to implement all models. The version of Torch is 1.0.0 (or above).

| | IMTNAS-DARTS | MTLSWS-DARTS | IMTNAS-PDARTS | MTLSWS-PDARTS | IMTNAS-PCDARTS | MTLSWS-PCDARTS |
|---|---|---|---|---|---|---|
| Runtime | 6272 | 5879 | 3466 | 2138 | 2541 | 2529 |

Table 19: Runtime (seconds per epoch) of architecture search

| | IMTNAS-DARTS | MTLSWS-DARTS | IMTNAS-PDARTS | MTLSWS-PDARTS | IMTNAS-PCDARTS | MTLSWS-PCDARTS |
|---|---|---|---|---|---|---|
| Num. of epochs | 50 | 50 | 50 | 50 | 50 | 100 |
| Validation accuracy (%) | 87.95 | 88.81 | 87.75 | 84.08 | 84.99 | 84.27 |

Table 20: Validation performance of architecture search on CIFAR-10

| | IMTNAS-DARTS | MTLSWS-DARTS | IMTNAS-PDARTS | MTLSWS-PDARTS | IMTNAS-PCDARTS | MTLSWS-PCDARTS |
|---|---|---|---|---|---|---|
| Num. of epochs | 50 | 50 | 50 | 50 | 50 | 100 |
| Validation accuracy (%) | 61.95 | 61.24 | 59.85 | 57.38 | 52.80 | 52.04 |

Table 21: Validation performance of architecture search on CIFAR-100

## G.2 For architecture evaluation

Table 22 and 23 show the runtime (seconds per epoch) of architecture evaluation.

| | Runtime |
|---|---|
| IMTNAS-DARTS | 137 |
| MTLSWS-DARTS | 92 |
| IMTNAS-PDARTS | 237 |
| MTLSWS-PDARTS | 199 |
| IMTNAS-PCDARTS | 212 |
| MTLSWS-PCDARTS | 205 |

Table 22: Runtime (seconds per epoch) of architecture evaluation on the CIFAR-10 dataset

| | Runtime |
|---|---|
| IMTNAS-DARTS | 122 |
| MTLSWS-DARTS | 95 |
| IMTNAS-PDARTS | 241 |
| MTLSWS-PDARTS | 199 |

Table 23: Runtime (seconds per epoch) of architecture evaluation on the CIFAR-100 dataset

## H  Full lists of hyperparameter settings

Table 24-32 show the hyperparameter settings used in different experiments of IMTNAS.

| Name | Value |
| --- | --- |
| Optimizer of architecture variables | Adam |
| Learning rate of architecture variables | 3e-4 |
| Weight decay of architecture variables | 0.001 |
| Optimizer of network weights | Momentum SGD |
| Learning rate of network weights | 0.025 |
| Minimum learning rate of network weights | 0.001 |
| Momentum of network weight | 0.9 |
| Weight decay of network weights | 3e-4 |
| Epochs | 50 |
| Batch size | 64 |
| Number of layers | 8 |
| Number of initial channels | 16 |
| Max gradient norm | 5.0 |

Table 24: Hyperparameter settings in IMTNAS-DARTS during architecture search. For MTLSWS-DARTS, most hyperparameter settings are the same as those in this table.

| Name | Value |
| --- | --- |
| Optimizer for architecture variables | Adam |
| Learning rate for architecture variables | 3e-4 |
| Weight decay for architecture variables | 0.001 |
| Optimizer for network weights | Momentum SGD |
| Learning rate for network weights | 0.025 |
| Minimum learning rate for network weights | 0.001 |
| Momentum for network weights | 0.9 |
| Weight decay for network weights | 3e-4 |
| Epochs | 50 |
| Batch size | 64 |
| Number of layers | 8 |
| Number of initial channels | 16 |
| Max gradient norm | 5.0 |

Table 25: Hyperparameter settings in IMTNAS-PDARTS during architecture search. For MTLSWS-PDARTS, most hyperparameter settings are the same as those in this table.

| Name | Value |
|---|---|
| Optimizer for architecture variables | Adam |
| Learning rate for architecture variables | 3e-4 |
| Weight decay for architecture variables | 0.001 |
| Optimizer for network weights | Momentum SGD |
| Learning rate for network weights | 0.025 |
| Minimum learning rate for network weights | 0.0 |
| Momentum for network weights | 0.9 |
| Weight decay for network weights | 3e-4 |
| Epochs | 50 |
| Batch size | 64 |
| Number of layers | 8 |
| Number of initial channels | 16 |
| Max gradient norm | 5.0 |

Table 26: Hyperparameter settings in IMTNAS-PCDARTS during architecture search. For MTLSWS-PCDARTS, most hyperparameter settings are the same as those in this table.

| Name | Value |
|---|---|
| Optimizer for architecture variables | Adam |
| Learning rate for architecture variables | 3e-4 |
| Weight decay for architecture variables | 0.001 |
| Optimizer for network weights | Momentum SGD |
| Learning rate for network weights | 0.025 |
| Minimum learning rate for network weights | 0.0 |
| Momentum for network weights | 0.9 |
| Weight decay for network weights | 3e-4 |
| Epochs | 50 |
| Batch size | 64 |
| Number of layers | 8 |
| Number of initial channels | 16 |
| Max gradient norm | 5.0 |

Table 27: Hyperparameter settings in IMTNAS-PRDARTS during architecture search.

| Name | Value |
|---|---|
| Optimizer | Momentum SGD |
| Learning rate | 0.025 |
| Epochs | 600 |
| Weight decay | 3e-4 |
| Weight momentum | 0.9 |
| Auxiliary weight | 0.4 |
| Cutout length | 16 |
| Path dropout probability | 0.2 |
| Batch size | 96 |
| Number of layers | 20 |
| Number of initial channels | 36 |
| Max gradient norm | 5.0 |

Table 28: Hyperparameter settings of IMTNAS-DARTS during architecture evaluation on CIFAR10 and CIFAR100. For MTLSWS-DARTS, most hyperparameter settings are the same as those in this table.

| Name | Value |
|---|---|
| Optimizer | Momentum SGD |
| Learning rate | 0.025 |
| Epochs | 600 |
| Weight decay | 3e-4 |
| Weight momentum | 0.9 |
| Auxiliary weight | 0.4 |
| Cutout length | 16 |
| Path dropout probability | 0.3 |
| Batch size | 96 |
| Number of layers | 20 |
| Number of initial channels | 36 |
| Max gradient norm | 5.0 |

Table 29: Hyperparameter settings of IMTNAS-PDARTS during architecture evaluation on CIFAR10 and CIFAR100. For MTLSWS-PDARTS, most hyperparameter settings are the same as those in this table.

| Name | Value |
|---|---|
| Optimizer | Momentum SGD |
| Learning rate | 0.025 |
| Epochs | 600 |
| Weight decay | 3e-4 |
| Weight momentum | 0.9 |
| Auxiliary weight | 0.4 |
| Cutout length | 16 |
| Path dropout probability | 0.3 |
| Batch size | 96 |
| Number of layers | 20 |
| Number of initial channels | 36 |
| Max gradient norm | 5.0 |

Table 30: Hyperparameter settings of IMTNAS-PCDARTS during architecture evaluation on CIFAR10 and CIFAR100. For MTLSWS-PCDARTS, most hyperparameter settings are the same as those in this table.

| Name | Value |
|---|---|
| Optimizer | Momentum SGD |
| Learning rate | 0.025 |
| Epochs | 600 |
| Weight decay | 3e-4 |
| Weight momentum | 0.9 |
| Auxiliary weight | 0.4 |
| Cutout length | 16 |
| Path dropout probability | 0.2 |
| Batch size | 96 |
| Number of layers | 20 |
| Number of initial channels | 36 |
| Max gradient norm | 5.0 |

Table 31: Hyperparameter settings of IMTNAS-PRDARTS during architecture evaluation on CIFAR10 and CIFAR100. For MTLSWS-PRDARTS, most hyperparameter settings are the same as those in this table.

| Name | Value |
|---|---|
| Optimizer | Momentum SGD |
| Learning rate | 0.5 |
| Learning rate decay gamma | 0.97 |
| Epochs | 250 |
| Weight decay | 3e-5 |
| Weight momentum | 0.9 |
| Auxiliary weight | 0.4 |
| Path dropout probability | 0 |
| Label smoothing epsilon | 0.1 |
| Batch size | 1024 |
| Number of layers | 14 |
| Number of initial channels | 48 |
| Max gradient norm | 5.0 |

Table 32: Hyperparameter settings of architecture evaluation on ImageNet

# I  Visualization of searched architectures

Figure 3-16 shows the visualizations of architectures searched in different experiments.

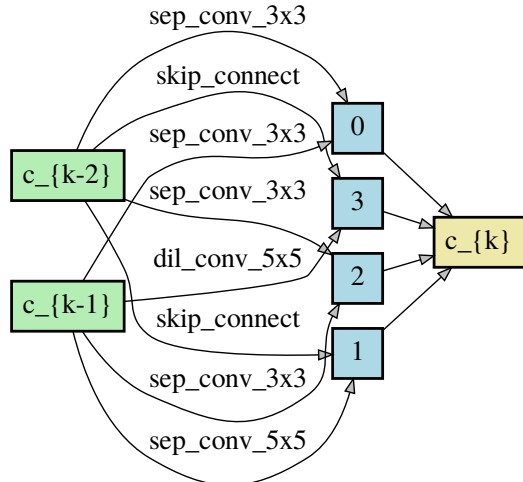

Figure 3: IMTNAS-DARTS normal cell.

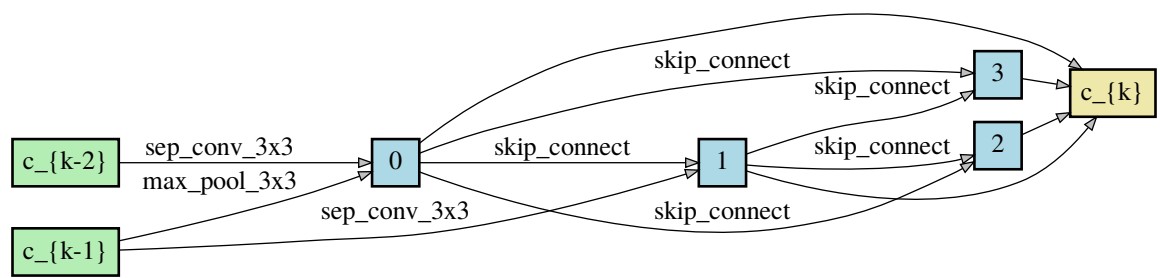

Figure 4: IMTNAS-DARTS reduce cell.

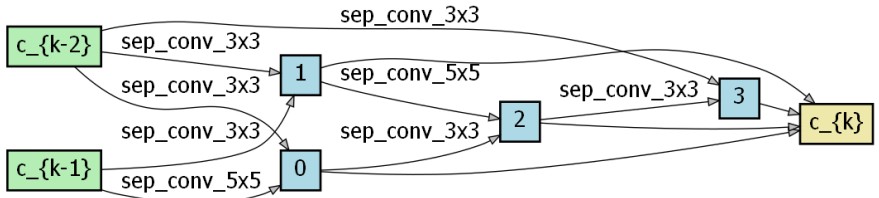

Figure 5: MTLSWS-DARTS normal cell.

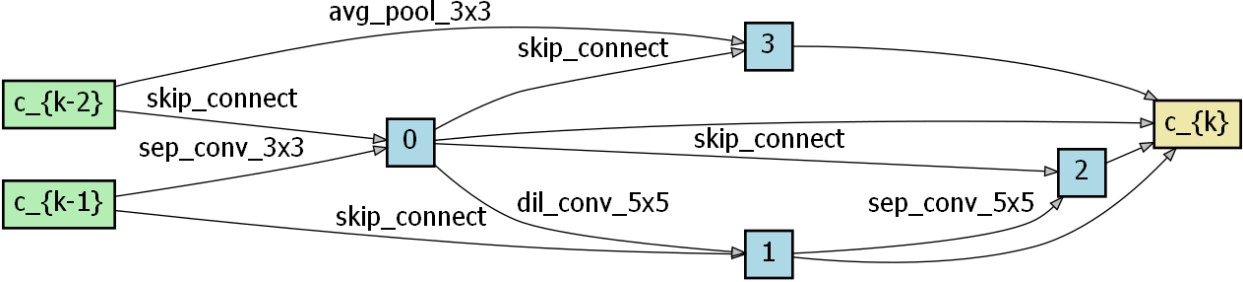

Figure 6: MTLSWS-DARTS reduce cell.

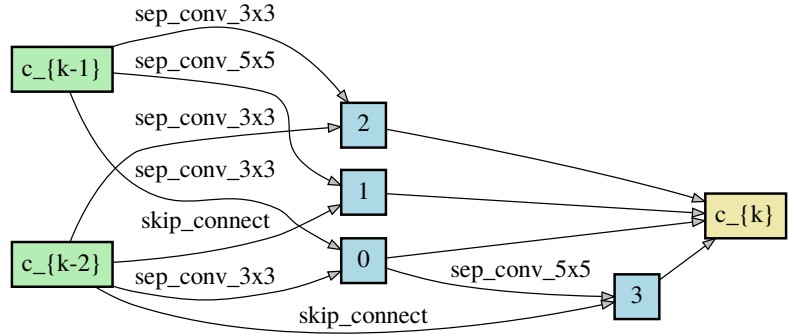

Figure 7: IMTNAS-PDARTS normal cell.

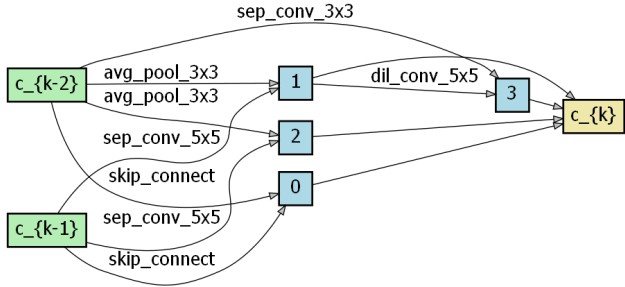

Figure 8: IMTNAS-PDARTS reduce cell.

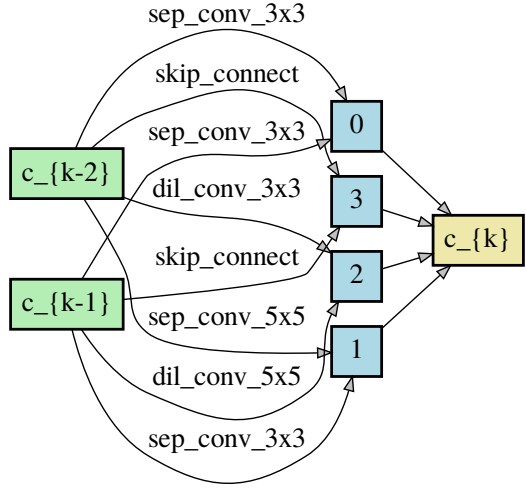

Figure 9: MTLSWS-PDARTS normal cell.

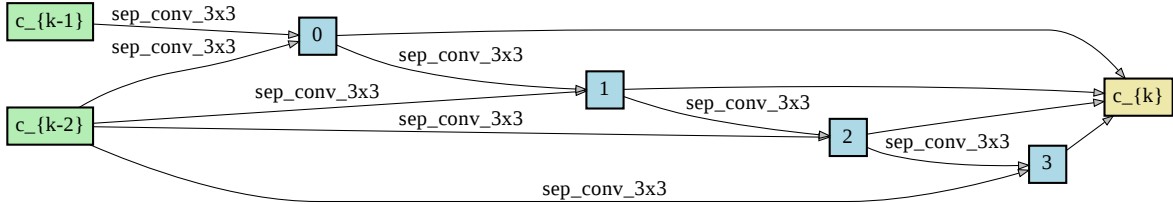

Figure 10: MTLSWS-PDARTS reduce cell.

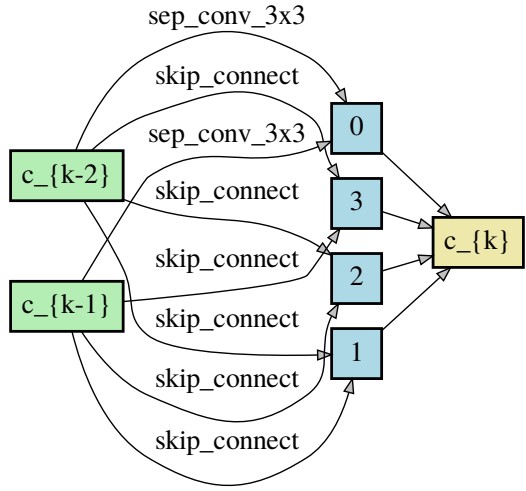

Figure 11: IMTNAS-PCDARTS normal cell.

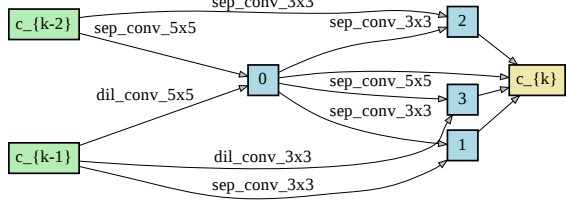

Figure 12: IMTNAS-PCDARTS reduce cell.

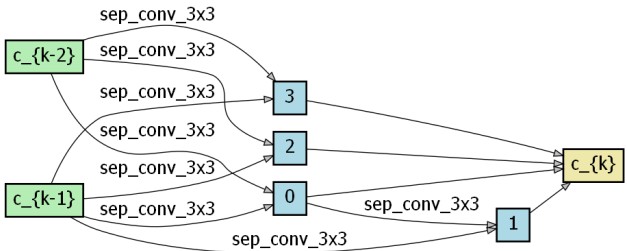

Figure 13: MTLSWS-PCDARTS normal cell.

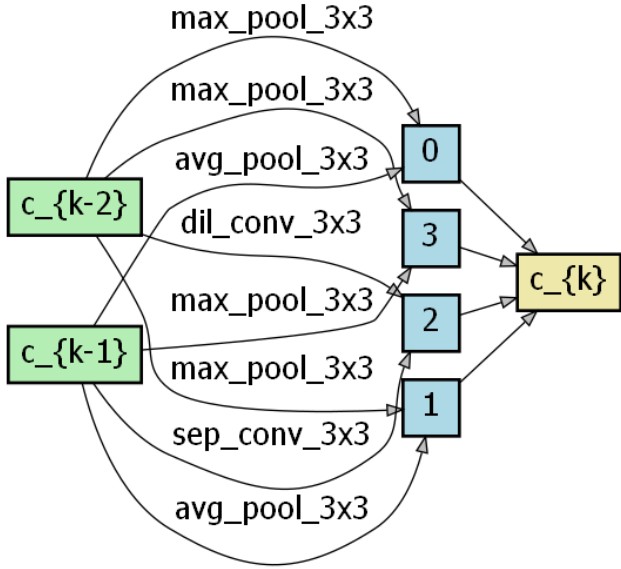

Figure 14: MTLSWS-PCDARTS reduce cell.

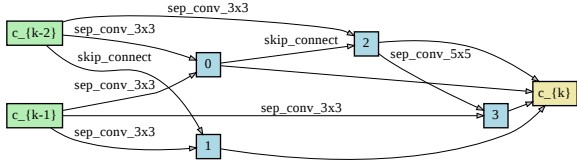

Figure 15: IMTNAS-PRDARTS normal cell.

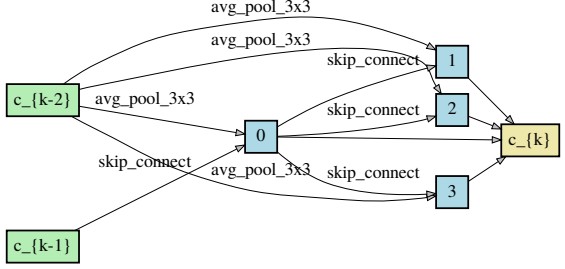

Figure 16: IMTNAS-PRDARTS reduce cell.

