# OpenReview forum: "Interleaving Multi-Task Neural Architecture Search"
_TMLR — Rejected by TMLR_

### Review · Reviewer_Dg1q · 2023-07-06

**Summary Of Contributions:**

The paper proposes a novel approach for neural architecture search for multi-task image classification/object detection networks. It does so by reformulating the problem as a multi-level optimization problem, and optimizes tasks in an interleaving fashion. It transfer knowledge across tasks based on a knowledge transfer method using distribution matching.

**Audience:**

Yes

**Broader Impact Concerns:**

No concerns

**Claims And Evidence:**

No

**Requested Changes:**

The paper requires a major revision of Section 3 in terms of clarity, readability and reproducibility. In its current form, too many details of the procedure remain unclear and not well motivated.
Also, it seems that the actual architecture optimization is just based on a sum of task losses, which renders the entire motivation of the paper pointless.
Experiments on more challenging multi-task benchmarks would strengthen the empirical support of the proposed method.


**Strengths And Weaknesses:**

Strength:
 * Multi-task neural architecture search is a relevant problem since with growing number of tasks, the design space for neural architectures becomes ever more complex.
 * Related works from multi-task learning, neural architectures search, multi-level optimization, and transfer learning are discussed.

Weaknesses:
 * The authors motivate that their method is beneficial as it does not require defining a weighted sum of losses across tasks as objective (which requires an explicit weighting of tasks). However, as of Equation (8), their NAS objective is just the sum of validation losses with weights being all 1 (and "normalized to have similar scales"). This renders the initial motivation pointless.
 * Key design choices of the method are not clearly motivated: why does knowledge transfer across tasks requires a distribution matching using MMD on augmented samples? (Section 3.1.2) What is the reason of not just copying weights? Actually, choosing $\tilde{W}_k^{(m)} =\tilde{W}_{k-1}^{(m)}(A)$ should minimize Equation 6 in its current form.
 * Related: how do ground-truth labels affect the middle stage's training of the encoder? In the current formulation, neither task labels nor head weights seem to affect the the choice of encoder weights as of Equation 6.
 * The paper lacks clarity in terms of presenting the method (Section 3) and would not allow reproduction. Providing detailed and clean pseudo-code for the procedure would be very helpful. Some exemplary parts which remain unclear to the reviewer:
    + as of Equation 8 and 9, there are still task-specific weights for the encoder at the end of the procedure. Shouldn't these be shared? The main benefit of a multi-task network is that computation of the encoder is shared across all tasks.
    + the lines in the "pseudo code" Equation 9 are strangely arranged; they need to be read bottom-to-top, which is confusing.
    + how does the hypernetwork interact with the multi-level optimization?
    + is NAS (Equation 8) done post-hoc once encoder and head weights are frozen? This is known to be suboptimal and interleaving NAS with weight updates should be considered. Specifically, since the method is called _interleaving_ multi-task neural architecture search, which would be a misnomer according to the current exposition of the method.
 * The experiments only use pairs of two tasks (classification+detection and CIFAR10+CIFAR100). It would be  relevant to check the approach also in settings with K > 2 tasks.

Minor comments:
 * I think the middle of Section 3.1.2 is not the right place for a lengthy introduction of MMD (if this is required at all)

---

### Review · Reviewer_FuWu · 2023-07-19

**Summary Of Contributions:**

The paper presents an approach to Multitask Neural Architecture Search.
They provide a multi-level optimization based solution that sequentially interleaves tasks. To transfer weights between interleaved tasks, they present an distribution matching objective.
Their approach is presented as efficient by appealing to the use of a hyper-network to amortize the memory cost of having unique weights per each task.

**Audience:**

No

**Broader Impact Concerns:**

No glaring ethical concerns noted.

**Claims And Evidence:**

No

**Requested Changes:**

Mostly I have asked quite a few clarification questions above.
Any updates to the paper should attempt to make it easier to understand the followup questions.

**Strengths And Weaknesses:**

# Strengths #
1. The paper presents comprehensive tables of comparisons to prior work
2. The paper conducts ablations to understand the effect of some of the components.

# Weaknesses #
I have several issues with this paper

 ## Motivation ##
The motivation for the paper does not seem strong. The cost of searching for optimal task weightings is presented as a main triumph over previous methods. However, there are several issues with this framing :
1. Table 4, which is meant to show the variation of the performance with task weights -- only highlights the fact that the default 1:1 per-task seems to work quite well without much tuning. In fact, several recent works have show that this default is actually a strong baseline when extensive experimentation is applied [1,2] -- though these are for non-architecture search MT, I would not be surprised that these results carry over.
2. Even though the explicit design choice of task weighting is removed, there is also an implicit weighting issue here --> the task ordering chosen. The paper does not go into detail about how the interleaving ordering is chosen or any ablations to validate that the ordering does not matter (at least that I could find -- please correct me if I missed this ). The task ordering can create implicit inequalities between tasks that are not dissimilar to those created from using different task weights during multitasking

## Methodological choices ##
There are some methodological choices that are not properly rationalized and seem more adhoc.
1. Why is distribution matching the choice used for transferring information between tasks ? Why would you expect to that distribution matching *across weights of different tasks* would be a useful objective. Note that all the tasks chosen have similar input spaces (detection vrs classification) .. (cifar10 vrs cifar100). I think this distribution matching strategy might actually be hurtful is more stark input space variation across tasks eg -- SVHN -> Cifar

2. Unsubstantiated claims and rationalizations — 3.1.2 — low order terms vrs higher order — why are higher order terms important ? What proof do you have that lower order isn’t sufficient ? This occurs in the first paragraph of section 3.1.2, the authors state "To address the limitations of existing methods, we propose a new knowledge transfer approach based on distribution matching, which is more flexible and can capture high-order (≥ 3)" --- also, isn't MMD first order in the moments of the distribution ?

3. Augmentations :    The datapoints used for the augmentations to get A_i and A_j …are they from task k-1 or from task k ?
    - Equation 6 is defined over all possible augmentation sets, over all possible pairs of data-points ? That’s a very expensive loss to compute -- am I missing something ?

4. Hypernetwork : The paper is lacking in the description of the hyper-network. Not much discussion of the hyper-network in the main text — what is its design ? How is the architecture encoded to feed it into the hyper-network ? why should the hyper-network depend on m — which round we are training on ?

# Experimental Design #

**“We randomly split the 118K training set into a new training and validation set with a ratio of 1:1.”**
    - What informed this choice ? Do the other methods compared to in the paper also use this same splits ?

From Table 13. DARTS+ has similar top 1 error , smaller final model size than your method but your method still takes 35x gpu days to train. !

**“We set the number of interleaving rounds to 2, with the following task order: C10,C100,C10,C100.”**
    - If you are using 1 step gradient descent as per the appendix — then it seems as if you’ll see only a handful of gradient descent steps
    - Hard for me to believe that the results are this good given this

In 4.3.1 you say you train for 40 epochs. What is an epoch in this setting ? How is it different from the number of rounds ?

# References #
[1] https://arxiv.org/abs/2201.04122

[2] https://arxiv.org/abs/2209.11379

---

### Review · Reviewer_DWqi · 2023-07-21

**Summary Of Contributions:**

This paper proposes an algorithm for multi-tasks neural architecture search (NAS), i.e., the paper aims to find an architecture which leads to good performances in multiple tasks. The paper takes an interleaving approach, and alternately optimizes the weights of each task while transferring knowledge from the preceding task. The algorithm is repeated for multiple $M$ iterations, and the architecture is optimized in the final stage after the weights of the tasks are optimized.

**Audience:**

Yes

**Broader Impact Concerns:**

The broader impact is properly discussed at the end of the main paper.

**Claims And Evidence:**

No

**Requested Changes:**

I've listed the requested changes/clarifications as weaknesses above.

**Strengths And Weaknesses:**

Strengths:
- The paper has proposed some interesting and intuitive ideas. It is interesting to interleave the different tasks and alternately optimize them. The idea of using augmentation data to encourage consistent learned representations between consecutive tasks is also an interesting idea.
- Comprehensive experiments are performed, and the results indeed demonstrate improvements over previous related methods on multi-task learning and multi-task NAS.
- The paper has performed a very comprehensive review of the related works.

Weaknesses:
- All experiments only involve 2 tasks, which I think may not be entirely realistic. Also, restricting to 2 tasks makes the problem much simpler and hence may not fully reveal the important insights about the algorithm. For example, at the bottom of page 11, the ablation study shows that the ordering of the tasks don't have significant impacts on the performance, however, I suspect that this observation could be because there are only two tasks, and if there is a large number of tasks, then different orderings of the tasks can cause important changes in the degree of information transfer. For example, consider two cases where (1) task 1 and task 2 are adjacent and (2) there are 10 intermediate tasks between task 1 and task 2, then there will be much less degree of information transfer between these two tasks in the latter case. So, I think it's important to perform more complicated experiments involving a larger number of tasks.
- Although the paper says that the focus of this work is NAS, I feel that the place where you optimize the model architecture may be only a minor component of your proposed algorithm. For example, if I understand correctly, the most important novelty of this work lies in the choice interleaving the tasks and the way to achieve information transfer through data augmentation; it is only after these procedures are completed will the architecture be optimized (Section 3.1.3). So, it seems to me that NAS may be only a minor part of the proposed algorithm, and if this is indeed the case, the paper needs to shift its focus through some revision to the writing.
- It appears that there are some incorrect hyperreferences in the Experiments section. For example, third line of Experiments section, should it instead of Eq. (6) instead of Eq. (7)? At the beginning of Section 4.3.2, should it be Table 2 instead of Table 14? And maybe the Table 15 at the beginning of Section 4.4.2 should be Table 3 instead.

---

### Review · Reviewer_7t1N · 2023-08-15

**Summary Of Contributions:**

This work proposes an interleaving Multi-task NAS method, named IMTNAS, which attempts to optimize each task in an interleaving loop. To transfer the knowledge from previous trained tasks, the authors resort to distribution matching technique. Specifically, the authors build several sets of data samples with different distribution, and encourage the model of new task to distinguish different sets of data in a similar manner of the model of previous task. After training the weights, the architecture is finally searched based on DARTS algorithm.

**Audience:**

Yes

**Claims And Evidence:**

Yes

**Requested Changes:**

More experiments should be conducted to demonstrate the efficacy of the proposed method.

**Strengths And Weaknesses:**

Strengths:
++ The motivation is quite interesting, and the proposed interleaving technique does not need to balance the loss weights. However, I am afraid that the order of tasks can affect the performance, please see weaknesses.

++ The experiments show great performance. However, I am afraid that the number of tasks is limited (only two tasks is tested in the experiments), please see weaknesses.

++ The proposed technique to transfer knowledge makes sense.

Weaknesses:
-- The order of tasks can affect the performance, especially when the tasks differ a lot. For example, consider four tasks: ImageNet, COCO, CIFAR10 and CIFAR100 in the improper orders, such as (ImageNet, Cifar10, Cifar100, COCO) or (ImageNet, COCO, CIFAR10, CIFAR100). In the first order, though knowledge of ImageNet will be transferred to CIFAR dataset, the knowledge of CIFAR can be useless for COCO.

-- The experimental results are insufficient to demonstrate the efficacy of the proposed method since only two tasks are tested. What about the performance of more tasks?

---

### Decision · Action_Editors · 2023-09-05

**Recommendation:** Reject

**Comment:**

In this paper, the authors designed a multi-task neural network architecture searching method, by iteratively transferring the learned network architecture to pre-ordered tasks, which bypasses the arbitrarily defined weights for losses combination in vanilla multi-task neural network architecture searching methods. The authors empirically tested the algorithms on the benchmarks, comparing with existing methods.

There are several major concerns raised by the reviewers:

- The algorithm design is adhoc, where the method design choice is overly complicated and without well-justified. (reviewer FuWu, Dg1q). Although the weights for multiple losses have been eliminated, the method introduces extra order in the algorithm.

- The experiments are not strongly enough to support the claim. As the reviewers suggested (DWqi, Dg1q), the method should be evaluated  in the learning setting with more than 2 tasks. Although the authors added in the rebuttal session, the comparison is not comprehensive.

- Reviewer Dg1q also pointed out the algorithm and hyperparameters are not specified clear, which introduces difficulties in reproducing.

**Audience:**

The paper should be interested to the neural network architecture search community.

**Claims And Evidence:**

The motivation and technical design choice are not fully supported, mainly lying in two-folds:

- The proposed method is not well-justified. The design choice of the algorithmic components are ad-hoc, and not properly rationalized.
(Reviewer FuWu, Dg1q)

- In the main text, the experiments are all conducted on two tasks setting, which is not sufficient to support the claim. Meanwhile, their results on 3 tasks in appendix does not demonstrate significant benefits comparing to two tasks setting, which is not aligned with the motivation (Reviewer DWqi, Dg1q)